# Customizing Sequence Generation with Multi-Task Dynamical Systems

## Abstract

Dynamical system models (including RNNs) often lack the ability to adapt the sequence generation or prediction to a given context, limiting their real-world application. In this paper we show that hierarchical *multi-task dynamical systems* (MTDSs) provide direct user control over sequence generation, via use of a latent code $\mathbf{z}$ that specifies the customization to the individual data sequence. This enables style transfer, interpolation and morphing within generated sequences. We show the MTDS can improve predictions via latent code interpolation, and avoid the long-term performance degradation of standard RNN approaches.

## 1 Introduction

Time series data often arise as a related 'family' of sequences, where certain characteristic differences exist between the sequences in a dataset. Examples include the style of handwritten text (Graves, 2013), the response of a patient to an anaesthetic (Bird et al., 2019), or the style of locomotion in motion capture (mocap) data (Ghosh et al., 2017). In this paper, we will consider how such variation may be modelled, and effectively controlled by an end user.

Such related data is often *pooled* to train a single dynamical system, despite the internal variation. For a simple model, such as a linear dynamical system (LDS), this will result in learning only an average effect. In contrast, a recurrent neural network (RNN) may model this variation, but in an implicit and opaque manner. Such a 'black-box' approach prohibits end-user control, and may suffer from mode drift, such as in Ghosh et al. (2017), where a generated mocap sequence performs an unprompted transition from walking to drinking. Some of these problems may be alleviated by appending 'context labels' to the inputs (see e.g. Goodfellow et al., 2016, §10.2.4) which describe the required customization. However, such labels are often unavailable, and the approach may fail to model the variation adequately even when they are.

To move beyond these approaches, we consider latent variable models, where a latent variable $\mathbf{z}$ characterizes each sequence. This may be seen as a form of multi-task learning (MTL, see Zhang & Yang, 2017), from which we derive the name multi-task dynamical system (MTDS), with each sequence treated as a task. A straightforward approach is to append the latent $\mathbf{z}$ to the inputs of the model, similarly to the 'context label' approach, thereby providing customization of the various bias (or offset) parameters of the model. A number of examples of this have been proposed recently, e.g. in Yingzhen & Mandt (2018) and Miladinović et al. (2019). Nevertheless, this 'bias customization' has limited expressiveness and is often unsuitable for customizing simple models.

In this paper we investigate a more powerful form of customization which modulates all the system and emission parameters. In this approach, the parameters of each task are constrained to lie on a learned low dimensional manifold, indexed by the latent $\mathbf{z}$. Our experiments show that this approach results in improved performance and/or greater data efficiency than existing approaches, as well as greater robustness to unfamiliar test inputs. Further, varying $\mathbf{z}$ can generate a continuum of models, allowing interpolation between sequence predictions (see Figure 1b for an example), and potentially *morphing* of sequence characteristics over time.

**Contributions**  In this paper we propose the MTDS, which goes beyond existing work by allowing full adaptation of *all* parameters of *general* dynamical systems via use of a learned nonlinear manifold. We show how the approach may be applied to various popular models, and provide general purpose

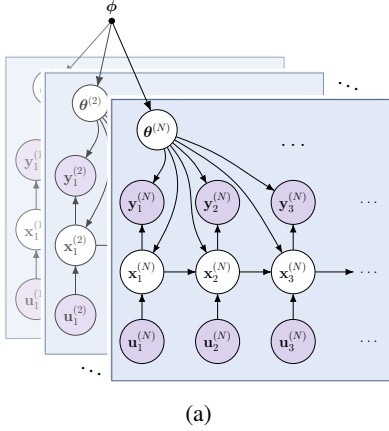
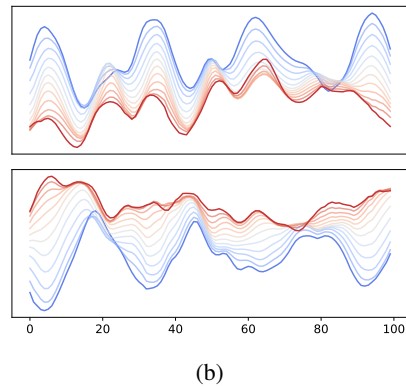

(a)

(b)

Figure 1: *(a)* Graphical Model of the multi-task dynamical system. *(b)* Right elbow joint during human locomotion measured in (top) vertical, and (bottom) horizontal directions, obtained varying **z** between 'childlike' (blue) and 'proud' (red) styles.

learning and inference algorithms. Our experimental studies use synthetic data (sum of two damped harmonic oscillators) and real-world human locomotion mocap data. We illuminate various properties of the MTDS formulation in our experiments, such as data efficiency, user control, and robustness to dataset shift, and show how these go beyond existing approaches to time series modelling. We finally utilize the increased user control in the context of mocap data to demonstrate *style morphing*.

To this end, we introduce the model in Section 2, giving examples and discussing the particular challenges in learning and inference. We discuss the relation to existing work in Section 3. Experimental setup and results are given in Section 4 with a conclusion in Section 5.

## 2 MULTI-TASK DYNAMICAL SYSTEMS

Consider a collection of input-output sequences $\mathcal{D} = \{Y^{(i)}, U^{(i)}\}_{i=1}^{N}$ with inputs $U^{(i)} = \{\mathbf{u}_1^{(i)}, \ldots, \mathbf{u}_{T_i}^{(i)}\}$ and outputs $Y^{(i)} = \{\mathbf{y}_1^{(i)}, \ldots, \mathbf{y}_{T_i}^{(i)}\}$, $i = 1, \ldots, N$, where $T_i$ denotes the length of sequence $i$. Each sequence $i$ is described by a different dynamical system, whose parameter $\boldsymbol{\theta}^{(i)}$ depends on the hierarchical latent variable $\mathbf{z}^{(i)} \in \mathcal{Z}$:

$$\boldsymbol{\theta}^{(i)} = \mathbf{h}_{\boldsymbol{\phi}}(\mathbf{z}^{(i)}), \quad \mathbf{z}^{(i)} \sim p(\mathbf{z}) \tag{1}$$

$$\mathbf{x}_t^{(i)} \sim p(\mathbf{x} \mid \mathbf{x}_{t-1}^{(i)}, \mathbf{u}_t^{(i)}, \boldsymbol{\theta}^{(i)}), \tag{2}$$

$$\mathbf{y}_t^{(i)} \sim p(\mathbf{y} \mid \mathbf{x}_t^{(i)}, \mathbf{u}_t^{(i)}, \boldsymbol{\theta}^{(i)}), \tag{3}$$

for $t = 1, \ldots, T_i$. The state variables $X^{(i)} = \{\mathbf{x}_1^{(i)}, \ldots, \mathbf{x}_{T_i}^{(i)}\}$, $\mathbf{x}_t \in \mathcal{X}$ follow the latent dynamics (2) starting from $\mathbf{x}_0 := \mathbf{0}$ (other choices of initial state are possible). See Figure 1a for a graphical model. In this paper we assume $\mathcal{Z} = \mathbb{R}^k$ which the vector-valued function $\mathbf{h}_{\boldsymbol{\phi}}(\cdot)$ transforms to conformable model parameters $\boldsymbol{\theta} \in \mathbb{R}^d, d \gg k$. Note that $\mathbf{h}_{\boldsymbol{\phi}}$ may keep some dimensions of $\boldsymbol{\theta}$ constant with respect to **z**. We call this a Multi-Task Dynamical System, going beyond the usage in Bird et al. (2019).

An MTDS model under this framework must specify three key quantities:

1. The base model (e.g. a LDS or RNN).

2. The nature of the interaction between the latent variable **z** and the parameter vectors $\boldsymbol{\theta}$ (e.g. if only the dynamics (eq. 2) depend on $\boldsymbol{\theta}$).

3. The choice of prior $p(\mathbf{z})$ and transformation $\mathbf{h}_{\boldsymbol{\phi}}$.

Where the use of a constant **z** over time is inappropriate, each sequence can be broken into segments of maximum length $L$. When $L$ is small enough, this effectively permits a time-varying **z**, as we will see in section 4.2.

## 2.1 EXAMPLES

In order to make the framework more concrete we will describe two general choices of the base model. In what follows we will write each parameter with a subscript $\mathbf{z}$ to denote dependence on $\mathbf{z}$ (e.g. $A_{\mathbf{z}} := A(\mathbf{z})$) to reduce notational clutter. The choice of $p(\mathbf{z})$ and $\mathbf{h}_{\phi}$ will depend on the application, but a fairly general choice is a deep latent Gaussian model (Kingma & Welling, 2014; Rezende et al., 2014). See section A.1.1 in the supplementary material for further discussion.

### 2.1.1 MULTI-TASK LINEAR DYNAMICAL SYSTEM

For a given $\mathbf{z}$, a multi-task linear dynamical system (MTLDS) can be described by:

$$\mathbf{x}_t = A_{\mathbf{z}}\mathbf{x}_{t-1} + B_{\mathbf{z}}\mathbf{u}_t + \mathbf{b}_{\mathbf{z}} + \mathbf{w}_t, \tag{4}$$

$$\mathbf{y}_t = C_{\mathbf{z}}\mathbf{x}_t + D_{\mathbf{z}}\mathbf{u}_t + \mathbf{d}_{\mathbf{z}} + \epsilon_t, \tag{5}$$

$\mathbf{w}_t \sim \mathcal{N}(\mathbf{0}, R_{\mathbf{z}}), \epsilon_t \sim \mathcal{N}(\mathbf{0}, S_{\mathbf{z}})$, with $\boldsymbol{\theta}_{\mathbf{z}} = \{A_{\mathbf{z}}, B_{\mathbf{z}}, \mathbf{b}_{\mathbf{z}}, C_{\mathbf{z}}, D_{\mathbf{z}}, \mathbf{d}_{\mathbf{z}}, R_{\mathbf{z}}, S_{\mathbf{z}}\} = \mathbf{h}_{\phi}(\mathbf{z})$. The parameterization of $\boldsymbol{\theta}_{\mathbf{z}}$ must satisfy the constraints of positive definite $R_{\mathbf{z}}$ and $S_{\mathbf{z}}$ and stable $A_{\mathbf{z}}$ (i.e. $\|A_{\mathbf{z}}\|_2 \leq 1$) for *all* $\mathbf{z}$, hence projection methods such as in Siddiqi et al. (2008) are not applicable. We choose an alternative formulation of the LDS, replacing the latent dynamics in eq. (4) by:

$$\mathbf{x}_t = \Sigma_{\mathbf{z}}Q_{\mathbf{z}}\mathbf{x}_{t-1} + B_{\mathbf{z}}\mathbf{u}_t + \mathbf{b}_{\mathbf{z}} + \mathbf{w}_t, \tag{6}$$

where $\Sigma_{\mathbf{z}}$ is a diagonal matrix and $Q_{\mathbf{z}}$ orthogonal with no loss of generality (proof in supp. mat.). Since $\|\Sigma_{\mathbf{z}}Q_{\mathbf{z}}\| \leq \|\Sigma_{\mathbf{z}}\|\|Q_{\mathbf{z}}\| = \|\Sigma_{\mathbf{z}}\|$, stability can be enforced e.g. by $\Sigma = \text{diag}\{\tanh(\boldsymbol{v})\}$ for some vector $\boldsymbol{v}$. For more details see section A.1.2 in the supplementary material.

### 2.1.2 MULTI-TASK RECURRENT NEURAL NETWORK

Due to the nonlinearity of an RNN, enforcing stability of $A_{\mathbf{z}}$ is not strictly required (see e.g. Miller & Hardt, 2019, §4.4), although bounding the spectral radius may be useful for learning (e.g. Pascanu et al., 2013). The dynamics of a multi-task RNN (MT-RNN) are described by:

$$\mathbf{x}_{t+1} = \tanh(A_{\mathbf{z}}\mathbf{x}_t + B_{\mathbf{z}}\mathbf{u}_t + \mathbf{b}_{\mathbf{z}}). \tag{7}$$

Combined with the emission model of eq. (5), we have $\boldsymbol{\theta}_{\mathbf{z}} = \{A_{\mathbf{z}}, B_{\mathbf{z}}, \mathbf{b}_{\mathbf{z}}, C_{\mathbf{z}}, D_{\mathbf{z}}, \mathbf{d}_{\mathbf{z}}\}$. If long-term dependencies are important we may consider an orthogonal transition matrix (parameterized as for the MTLDS) to create a multi-task version of the Orthogonal RNN (ORNN, Helfrich et al., 2018).

## 2.2 LEARNING

The parameters $\phi$ of an MTDS can be learned from a dataset $\mathcal{D} := \{Y^{(i)}, U^{(i)}\}_{i=1}^N$ via maximum marginal likelihood: $\phi^* = \arg\max_{\phi} \sum_{i=1}^N \log p(Y^{(i)} \mid U^{(i)}, \phi)$, where

$$\log p(Y \mid U, \phi) = \log \int_{\mathcal{Z}} p(Y|U, \mathbf{h}_{\phi}(\mathbf{z})) \, p(\mathbf{z}) \, \mathrm{d}\mathbf{z}. \tag{8}$$

The first term in the integrand, $p(Y|U, \mathbf{h}_{\phi}(\mathbf{z})) = \int_{\mathcal{X}^T} p(Y|X, U, \mathbf{h}_{\phi}(\mathbf{z})) \, p(X|U, \mathbf{h}_{\phi}(\mathbf{z})) \, \mathrm{d}X$ is generally intractable for stochastic dynamics (with notable exceptions of discrete and linear-Gaussian models). A common approach is to use a variational *evidence lower bound* (ELBO) see e.g. Fraccaro et al. (2016); Goyal et al. (2017); Miladinović et al. (2019) or a *Monte Carlo objective* (MCO) e.g. Maddison et al. (2017); Le et al. (2018); Naesseth et al. (2018). For clarity of exposition we only consider models with deterministic state, which extends to the MTRNN and MTLDS above (in the case $\mathbf{w}_t = \mathbf{0}$ for all $t$). This also avoids the interaction effect with the choice of approximate marginalization over $X$.

Equation (8) also cannot be computed in closed form in general, and so we resort to approximate learning. A natural choice is via the ELBO (see an alternative MCO approach for unsupervised tasks in Section A.1.3, supp. mat.). We write the ELBO of eq. (8) as:

$$\mathcal{L}(Y; \phi, \boldsymbol{\lambda}) = \mathbb{E}_{q_{\boldsymbol{\lambda}}(\mathbf{z}|Y,U)}\left[\log p(Y|U, \mathbf{h}_{\phi}(\mathbf{z}))\right] - D_{\mathrm{KL}}\left(q_{\boldsymbol{\lambda}}(\mathbf{z}|Y,U)\|p(\mathbf{z})\right), \tag{9}$$

where $D_{\mathrm{KL}}$ is the Kullback-Leibler divergence and $q_{\boldsymbol{\lambda}}(\mathbf{z}|Y,U)$ an approximate posterior for $\mathbf{z}$. We can now optimize a lower bound of the marginal likelihood via $\arg\max_{\phi,\boldsymbol{\lambda}} \sum_{i=1}^N \mathcal{L}(Y^{(i)}; \phi, \boldsymbol{\lambda})$,

where low variance unbiased gradients of eq. (9) are available via reparameterization (Kingma & Welling, 2014; Rezende et al., 2014) and minibatches of size $N_{\text{batch}} < N$. Optimization via Adam (Kingma & Ba, 2014) proved adequate in our experiments without resorting to any specialized RNN optimization tricks. A standard choice of $q_{\boldsymbol{\lambda}}(\mathbf{z}|Y, U)$ is $\mathcal{N}(\boldsymbol{\mu}_{\boldsymbol{\lambda}}(Y, U), \mathbf{s}_{\boldsymbol{\lambda}}(Y, U))$, where $\boldsymbol{\mu}_{\boldsymbol{\lambda}}, \mathbf{s}_{\boldsymbol{\lambda}}$ are inference networks (e.g. Fabius & van Amersfoort, 2015).

It can be difficult to learn a sensible latent representation if the base model is a powerful RNN. When each output can be identified unambiguously via the inputs preceding it, a larger ELBO can be obtained by the RNN learning the relationship without using the latent variable (see e.g. Chen et al., 2017). A useful heuristic for avoiding such optima is KL annealing (e.g. Bowman et al., 2016). In our experiments we perform an initial optimization without the KL penalty (second term in eq. 9), initializing $\mathbf{s}_{\boldsymbol{\lambda}}(Y, U)$ to a small constant value.

## 2.3 INFERENCE

For an unseen test sequence $\{Y', U'\}$, the posterior predictive distribution is $p(\mathbf{y}'_{t+1:T} \,|\, \mathbf{y}'_{1:t}, \mathbf{u}'_{1:T}) = \int_{\mathcal{Z}} p(\mathbf{y}'_{t+1:T} \,|\, \mathbf{u}'_{1:T}, \mathbf{z}) \, p(\mathbf{z} \,|\, \mathbf{y}'_{1:t}, \mathbf{u}'_{1:t}) \, \mathrm{d}\mathbf{z}$, usually estimated via Monte Carlo. The key quantity is the posterior over $\mathbf{z}$, which may be approximated by the inference networks $\boldsymbol{\mu}_{\boldsymbol{\lambda}}, \mathbf{s}_{\boldsymbol{\lambda}}$. However, for novel test sequences, the inference networks may perform poorly and standard approximate inference techniques may be preferred. For further discussion and a description of our inference approach, see sections A.1.4, A.1.5 in the supp. mat. We note that $\mathbf{z}$ may not require inference, for instance by using the posterior of a sequence in the the training set. This may be useful for artistic control, style transfer, embedding domain knowledge or overriding misleading observations. We also note that the latent code can be varied during the state rollout to simulate a task which varies over time.

## 3 RELATED WORK

A number of dynamical models following the 'bias customization' approach have been proposed recently. Miladinović et al. (2019) and Hsu et al. (2017) propose models where the biases of an LSTM cell depend on a (hierarchical) latent variable. Yingzhen & Mandt (2018) propose a dynamical system where the latent dynamics are concatenated with a time-constant latent variable. In contrast, our MTDS model performs full parameter customization, and this on both the dynamics and emission distributions. A number of other proposals may be considered specialized applications of the MTDS. Bird et al. (2019) use a small deterministic nonlinear dynamical system for the base model whose parameters depend on $\mathbf{z}$ using a nonlinear factor analysis structure. Spieckermann et al. (2015) use a small RNN as the base model, where the transition matrix depends on $\mathbf{z}$ via multilinear decomposition. Lin et al. (2019) use a small stochastic nonlinear dynamical system with $\mathbf{h}_\phi$ a set of parameter vectors chosen discretely (or in convex combination) via $\mathbf{z}$.

Controlling and customizing sequence prediction has received much attention in the case of video data. As in the MTDS, these approaches learn features that are constant (or slowly varying) within a subsequence. Denton & Birodkar (2017) and Villegas et al. (2017) propose methods for disentangling time-varying and static features, but do not provide a useful density over the latter, nor an obvious way to customize the underlying dynamics. Tulyakov et al. (2018) use a GAN architecture where $\mathbf{z}$ factorizes into content and motion components. Hsieh et al. (2018) force a parts-based decomposition of the scene before inferring the latent content $\mathbf{z}$. However, as before, the dynamic evolution cannot be easily customized with these methods.

Hierarchical approaches for dynamical systems with *time-varying* parameters are proposed in Luttinen et al. (2014) (corresponding to non-stationary assumptions) and in Karl et al. (2017) (for the purposes of local LDS approximation). These models, like the MTDS can adapt all the parameters, but are linear and correspond to single task problems. Rangapuram et al. (2018) predict the parameters of simple time-varying LDS models directly via an RNN. While this is a multi-task problem, it is assumed that all necessary variation can be inferred from the inputs $U$.

Multi-task GPs are commonly used for sequence prediction. Examples include those in Osborne et al. (2008); Titsias & Lázaro-Gredilla (2011); Álvarez et al. (2012); Roberts et al. (2013). MTGPs however can only be *linear* combinations of (a small number of) latent functions, further, predictions depend critically upon often unknown mean functions, and inputs are not easily integrated. Note that an MTDS with no inputs, an LDS base model, a linear-Gaussian prior over the emission parameters

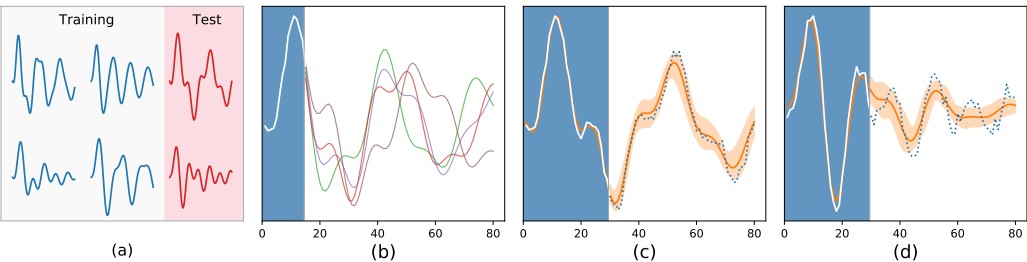

Figure 2: *(a)* Example DHO data for $N = 4$ training sequences and two test sequences. *(b)* Samples from the MTLDS trained on the 4 sequences in 2a, conditioned on the shaded region of test sequence 1 up to $t = 15$. *(c)* Posterior predictive density of the same data and model at $t = 30$. Posterior mean and 95% C.I. shown in orange, true values shown in dotted blue. *(d)* Posterior mean and C.I. for the second test sequence.

and fixed dynamics is an MTGP. In contrast to MTGPs, an MTDS allows much greater flexibility of the *dynamic* variation. The MT GP dynamical system of Korkinof & Demiris (2017) mitigates some of these limitations, but it retains a simple linear combination of latent dynamics with a Gaussian density over the combination.

Style transfer has been widely explored for mocap data e.g. Hsu et al. (2005); Min et al. (2010); Xia et al. (2015); Holden et al. (2017a), but in most previous work, generating sequences conditioned on a trajectory is not possible. For recurrent sequence generation, Fragkiadaki et al. (2015) proposed multilayer LSTM approaches with an encoding and decoding network. Martinez et al. (2017), introduced the idea of *open loop* or 'sampled' training (cf. Bengio et al., 2015) in order to avoid recurrent models converging quickly towards a mean pose. However, style transfer is unavailable with these methods. A phase-varying nonlinear autoregressive approach was introduced in Holden et al. (2017b) which was extended in Mason et al. (2018) to provide style transfer, but no quantitative results are available. We are not aware of any existing work which allows style interpolation.

## 4 EXPERIMENTS

We investigate the performance of the MTDS on two datasets. The first experiment investigates the performance of the MTDS on synthetic data generated by linear superposition of damped harmonic oscillation (DHO). The second experiment considers real-world mocap data for human locomotion.

### 4.1 DAMPED HARMONIC OSCILLATION

**Data** The generative model for $J$ oscillators with constant amplitudes $\gamma$ and variable frequency and decay factors, $\omega \in \mathbb{R}_+$, $\rho \in (0, 1]$ is $y_t^{(i)} = \sum_{j=1}^{J} \gamma_j (\rho_j^{(i)})^t \sin(\omega_j^{(i)} t) + \epsilon_t^{(i)}$, $t = 1, \ldots, 80$ for tasks $i = 1, 2, \ldots, N$. The emission noise is distributed iid as $\epsilon_t^{(i)} \sim \mathcal{N}(0, 0.05^2)$ and the amplitudes $\gamma = [1.0, -0.5]$. We use $J = 2$ oscillators with the 'sequence family' thus parameterized by a 4-d space. Details of the generating distribution for $\omega$ and $\rho$ are given in supplementary material sec. A.2.1. See Figure 2(a) for example traces.

**Model** We model the DHO data using an MTLDS with deterministic state $\mathcal{X} = \mathbb{R}^4$ and a $k = 4$ latent variable $\mathbf{z}$. All LDS parameters were adapted via the latent $\mathbf{z}$ except $D := 0$ and the emission variance $s^2$, which was learned. For optimization, we use the MCO algorithm of section A.1.3. This can obtain a tighter bound than the ELBO, and is useful to investigate convergence to the true model over increasing $N$. We contrast this with a bias customization approach (e.g. Miladinović et al., 2019), implemented similarly, but such that only the parameter $\mathbf{b}$ (eq. 6) depends on $\mathbf{z}$. We also train a Pooled LDS, which is the standard approach, using the same parameters for all tasks, and a single-task (STL) LDS which is learned from scratch using Bayesian inference over all parameters for each task. The Pooled-LDS was initialized using spectral methods (see Van Overschee & De Moor,

| | DHO RMSE | | | | | |
| | $N = 4^*$ | | $N = 16$ | | $N = 128$ | |
| **Model** | $t = 20$ | $t = 40$ | $t = 20$ | $t = 40$ | $t = 20$ | $t = 40$ |
| --- | --- | --- | --- | --- | --- | --- |
| Pooled LDS | 0.37 | 0.31 | 0.36 | 0.31 | 0.36 | 0.31 |
| STL LDS | 0.36 | 0.11 | - | - | - | - |
| LDS MT Bias | 0.41 | 0.33 | 0.37 | 0.30 | 0.33 | 0.27 |
| LDS MT Full | 0.18 | 0.12 | **0.11** | **0.07** | **0.09** | **0.06** |

Table 1: DHO results using LDS models, with training set size $N$ shown. Predictive RMSE after $t = 20, 40$. (*) The STL model trains from scratch, and hence on $N = 0$ sequences.

2012) and then fine tuned using Adam. The STL model requires no training as it is inferred directly on test data. More details are given in section A.2 in the supplementary material.

**Evaluation**   We assess how quickly and effectively the models can adapt to novel test sequences with a training set size of $N = 2^1, 2^2, \ldots, 2^7$. (The STL approach effectively uses $N = 0$.) The test set comprises 20 additional sequences drawn from the generating distribution. For an initial subsequence $y_{1:t}$, we estimate the predictive posterior $p(y_{t+1:T} | y_{1:t})$ for various $t$ and assess the predictions via root mean squared error (RMSE) and negative log likelihood (NLL). For MTL we use the Monte Carlo inference method described in supp. mat. A.1.5 and for STL we use Hamiltonian Monte Carlo (NUTS, Hoffman & Gelman, 2014). Each experiment is repeated 10 times to estimate sampling variance.

**Results**   The results, shown in Table 1 and supp. mat. section A.2.2, show substantial advantage of using the MTLDS ('MT Full') over single-task or pooled approaches. The MTLDS consistently outperforms the Pooled-LDS for all training sizes $N \geq 4$. Merely performing bias customization ('MT Bias') is insufficient to perform much better than a pooled approach. An example of MTLDS test time prediction is shown in Figure 2, with Figures 2c and 2d demonstrating effective generalization from the $N = 4$ training examples (Figure 2a). Even after 40 observations, the STL approach (which is capable of fitting each sequence exactly) does not significantly outperform the $N = 4$ MTLDS. Furthermore, the runtime was approx. 1000 times longer since STL inference is higher dimensional and poorly conditioned, and requires a more expensive algorithm. Note that with a larger training set size of $N = 128$, the MLTDS approaches the likelihood of the true model (Figure 7, supp. mat.).

## 4.2   MOCAP DATA

**Data**   The dataset consists of 31 sequences from Mason et al. (2018) (ca. 2000 frames average at 30fps) in 8 styles: angry, childlike, depressed, neutral, old, proud, sexy, strutting. In this case the family of possible sequences corresponds to differing walking styles. Each observation represents a 21-joint skeleton in a Lagrangian frame, $\mathbf{y}_t \in \mathbb{R}^{64}$ where the root movement is represented by a smoothed component and its remainder. Following Mason et al. (2018) we represent joints by their spatial position rather than their rotation. We also provide inputs that an animator may wish to control: the root trajectory over the next second, the gait cycle and a boolean value determining whether the skeleton turns around the inside or outside of a corner. See section A.3.1 in the supplementary materials for more details.

**Model**   We use a recurrent 2-layer base model where the first hidden layer is a 1024 unit GRU (Cho et al., 2014) and the second hidden layer is a 128 unit standard RNN, follwed by a linear decoding layer. The first-layer GRU does not vary with $\mathbf{z}$, i.e. it learns a shared representation of the input sequence across all $i$. Explicitly, omitting index $i$, the model for a given $\mathbf{z}$ is:

$$[\boldsymbol{\psi}_2, C, \mathbf{d}] = \mathbf{h}_\phi(\mathbf{z}), \tag{10}$$
$$\mathbf{x}_{1,t} = \text{GRUCell}_{1024}\{\text{state}= \mathbf{x}_{1,t-1}, \text{ input}= \mathbf{u}_t; \ \boldsymbol{\psi}_1\}, \tag{11}$$
$$\mathbf{x}_{2,t} = \text{RNNCell}_{128}\{\text{state}= \mathbf{x}_{2,t-1}, \text{ input}= H\mathbf{x}_{1,t-1}; \ \boldsymbol{\psi}_2\}, \tag{12}$$
$$\hat{\mathbf{y}}_t = C\mathbf{x}_{2,t} + \mathbf{d}, \tag{13}$$

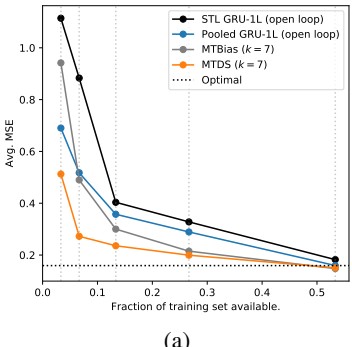 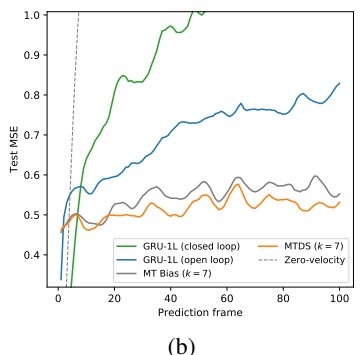

| Target | MT Bias | MTDS |
|--------|---------|------|
| Angry  | 0.78    | **0.86** |
| Child  | 0.59    | **0.95** |
| Depr.  | 0.65    | **0.81** |
| Neut.  | 0.79    | **0.93** |
| Old    | 0.55    | **0.86** |
| Proud  | 0.71    | **0.82** |
| Sexy   | 0.55    | **0.90** |
| Strut  | 0.71    | **0.92** |

(a)                              (b)                              (c)

Figure 3: *(a)* Mocap Experiment 1: out-of-sample MSE by % of training set seen. The performance achieved by the GRU models for the entire training set is shown as the 'Optimal' dashed line. *(b)* Experiment 2: MSE performance (avg over LOO), truncated for clarity, see supp. mat. for the full range. *(c)* Experiment 3: Classifier probability of target style averaged over all input styles.

for $t = 1, \ldots, T$. The parameters are $\boldsymbol{\theta} = \{\boldsymbol{\psi}_1, \boldsymbol{\psi}_2, H, C, \mathbf{d}\}$ where $\boldsymbol{\psi}_1$ and $H$ are constant wrt. $\mathbf{z}$. The matrix $H \in \mathbb{R}^{\ell \times 1024}$ ($\ell < 1024$) induces a bottleneck between layers, forcing $\mathbf{z}$ to explain more of the variance. For our experiments, a small $\ell$ can be used (we use $\ell = 24$). The first layer GRU uses 1024 units since it was observed experimentally to produce smoother animations than smaller networks. The second layer does not use a gated architecture, as gates appear to learn style inference more easily, and result in less use of $\mathbf{z}$.

For learning, each sequence was broken into overlapping segments of length 64 (approx. two second intervals), which allows $\mathbf{z}$ to vary across a sequence. We learn the model using an open-loop objective, i.e. the $\mathbf{y}_t$ are not appended to the inputs. This forces the model to recover from its mistakes as in Martinez et al. (2017), although unlike these approaches, we do not append predictions to the inputs either. Our rationale is that the state captures the same information as the predictions, and while previous approaches required observations $\mathbf{y}_{1:\tau}$ as inputs to seed the state, we can use the latent $\mathbf{z}$. The model was optimized using the variational procedure in section 2.2, where a slower learning rate (by a factor of 10-50) for the first layer parameters (i.e. $\boldsymbol{\psi}_1, H$) usually resulted in a more descriptive $\mathbf{z}$. We also found that standard variational inference for each $\mathbf{z}^{(i)}$ worked better in general than using amortized inference.

For comparison, we implement a bias customization model ('MTBias') via a deterministic state version of Miladinović et al., 2019, which follows eqs. (10)-(13) but only the RNN bias in eq. (12) is a function of $\mathbf{z}$. We also implement a 1-layer and 2-layer GRU without the multi-task apparatus, which serves both as an ablation test and a competitor model (Martinez et al., 2017) on the new dataset. Style inference is performed with the same network given an initial seed sequence $\mathbf{y}_{1:\tau}$. We train these in closed-loop (i.e. traditional next step 'teacher forcing' criterion) and open-loop (Martinez et al., 2017) settings. For baselines, we use constant predictions of (*i*) the training set mean and (*ii*) the last observed frame of the seed sequence ('zero-velocity' prediction).

**Experiment 1** We test the data efficiency of the MTDS by training the models on subsets of the original dataset. Besides the models described above, 8 'single-task' versions of the GRU models are trained which only see data for a single style. We use six training sets of approximate size $2^8, 2^9, 2^{10}, 2^{11}, 2^{12}, 2^{13}$ frames per style, where sampling is stratified carefully across all styles, and major variations thereof. For all experiments, the model fit (MSE) is calculated from the same 32 held out sequences (each of length 64). The results are shown in Figure 3a. As expected, the MTDS, MTBias and Pooled models obtain 'multi-task' gains over STL approaches for small datasets. However, the MTDS demonstrates much greater data efficiency, achieving close to the minimum error with only 7% of the dataset. The MTBias model requires more than twice this amount to obtain the same performance, and the Pooled model requires more than four times this amount. More details, as with all mocap experiments, can be found in supp. mat. section A.2.3.

**Experiment 2** We investigate how well the MTDS can generalize to novel sequence styles via use of a leave-one-out (LOO) setup, similar to transfer learning. For each test style, a model is trained on the other 7 styles in the training set, and hence encounters novel sequence characteristics at test time. We average the test error over the LOO folds as well as 32 different starting locations on each test sequence. The results are given in Figure 3b. We see that while the competitor (pooled) models perform well initially, they usually degrade quickly (worse for closed-loop models). In contrast, the multi-task models finds a better customization which evidences no obvious worsening over the predictive interval. Unlike pooled-RNNs, the MTDS and MTBias models can firstly perform correct inference of their customization, and secondly can 'remember' it over long intervals. We note that all models struggle to customize the arms effectively, since their test motions are often entirely novel. Customization to the legs and trunk is easier since less extrapolation is required (see animation videos linked in section A.4.1).

**Experiment 3** We investigate the control available in the latent $\mathbf{z}$ by performing style transfer. For various inputs $U^{(s_1)}$ from each source style $s_1$, we generate predictions from the model using target style $s_2$, encoded by $\mathbf{z}^{(s_2)}$. We use a classifier with multinomial outputs, trained on the 8 styles of the training set, to test whether the target style $s_2$ can be recognized from the data generated by the MTDS. Figure 3c gives the classifier 'probability' for each *target* style $s_2$, averaged over all the inputs $\{U^{(s_1)} : s_1 \neq s_2\}$. Successful style transfer should result in a the classifier assigning a high probability to the target style. These results suggest that the prediction style can be well controlled by $\mathbf{z}^{(s_2)}$ in the case of the full MTDS, but the MTBias demonstrates reduced control for some (source, target) pairs. See the videos linked in section A.4.1 for examples, and sec. A.2.3 for more details.

**Qualitative investigation** Qualitatively, the MTDS appears to learn a sensible manifold of walking styles, which we assess through visualization of the latent space. A $k = 2$ latent embedding can be seen in Figure 4 where the $\mathbf{z}^{(i)}$ for each training segment $i$ is coloured by the true style label. Some example motions are plotted in the figure. The MTDS embedding broadly respects the style label, but learns a more nuanced representation, splitting some labels into multiple clusters and coalescing others. These appear broadly valid, e.g. the 'proud' style contains both marching and arm-waving, with the latter similar to an arm-waving motion in the 'childlike' style. This highlights the limitation of relying on task labels. Visualizations such as Fig. 1b indicate that smooth style interpolation is available via interpolation in latent space. We take advantage of this in the animations (linked from sec. A.4.1) by morphing styles dynamically.

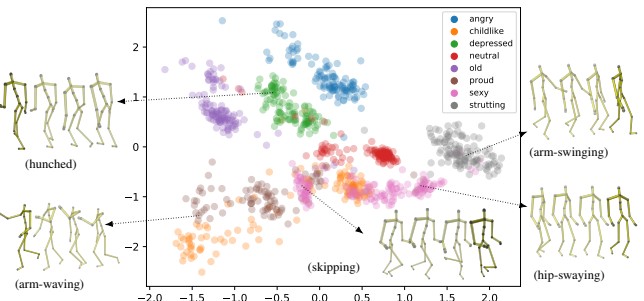

Figure 4: $k = 2$ mean embedding of each sequence segment, coloured by its (unseen) task label.

## 5 CONCLUSION

In this work we have shown how to extend dynamical systems with a general-purpose hierarchical structure for multi-task learning. Our MTDS framework performs customization at the level of all parameters, not just the biases, and adapts all parameters for general classes of dynamical systems. We have seen that the latent code can learn a fine-grained embedding of sequence variation and can be used to modulate predictions.

Clearly good predictive performance for sequences requires task inference, whether implicit or explicit. There are three advantages of making this inference explicit. Firstly, it enhances control over predictions. This might be used by animators to control the style of predictions for mocap models, or to express domain knowledge, such as ensuring certain sequences evolve similarly. Secondly, it can improve generalization from small datasets since task interpolation is available out-of-the-box. Thirdly, it can be more robust against changes in distribution at test time than a pooled model:

standard inference techniques or human supervision can guard against poor performance of the implicit inference.

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

# A  SUPPLEMENTARY MATERIAL

## A.1  MODEL

In this section we elaborate on some aspects of the MTDS model which were omitted from the main text. We discuss the choice of the prior in section A.1.1, give further details of the MTLDS parameterization in section A.1.2, provide an alternative learning algorithm in section A.1.3 (which is especially suited for unsupervised models), and discuss choices of inference algorithm in sections A.1.4-A.1.5.

### A.1.1  CHOICE OF PRIOR

The usual choice of $p(\mathbf{z})$ in latent variable models following Kingma & Welling (2014); Rezende et al. (2014) is a unit Gaussian $p(\mathbf{z}) = \mathcal{N}(\mathbf{0}, I)$. This choice allows simple sampling schemes, and straight-forward posterior approximations. It is also a useful choice for interpolation, since it allows continuous deformation of its outputs. An alternative choice might be a uniform distribution over a compact set, however posterior approximation is more challenging, see Svénsen (1998) for one approach.

Sensible default choices for $\mathbf{h}_\phi$ include affine operators and multilayer perceptrons (MLPs). However, when the parameter space $\mathbb{R}^d$ is large, it may be infeasible to predict $d$ outputs from an MLP. Consider an RNN with 100k parameters. If an MLP has $m_{L-1} = 300$ units in the final hidden layer, the expansion to the RNN parameters in the final layer will require $30 \times 10^6$ parameters alone. A practical approach is to use a low rank matrix for this transformation, equivalent to adding an extra linear layer of size $m_L$ where we must have $m_L \ll m_{L-1}$ to reduce the parameterization sufficiently. Since we will typically need $m_L$ to be $\mathcal{O}(10)$, we are restricting the parameter manifold of $\boldsymbol{\theta}$ to lie in a low dimensional subspace.

Since MLP approaches with a large base model will then usually have a restricted final layer, are there any advantages over a simple linear-Gaussian model for the prior $p(\mathbf{z})$ and $\mathbf{h}_\phi$? There may indeed be many situations where this simpler model is reasonable. However, we note some advantages of the MLP approach:

1. The MLP parameterization can shift the density in parameter space to more appropriate regions via nonlinear transformation.

2. A linear space of recurrent model parameters can yield highly non-linear changes even to simple dynamical systems (see e.g. the bifurcations in §8 of Strogatz, 2018). We speculate it might be advantageous to curve the manifold to avoid such phenomena.

3. More expressive choices may help utilization of the latent space (e.g. Chen et al., 2017). This may in fact motivate moving beyond a simple MLP for the $\mathbf{h}_\phi$.

### A.1.2  MULTI-TASK LINEAR DYNAMICAL SYSTEM PARAMETERIZATION

The matrices $A$, $B$, $R$, $S$ of the MTLDS can benefit from specific parameterizations, which we will discuss in turn.

**Degeneracy of LDS.**  It will be useful to begin with the well-known over-parameterization of linear dynamical systems. The hidden dynamics of a LDS can be transformed by any invertible matrix $G$ while retaining the same distribution over the emissions $Y$. This follows essentially because the basis used to represent $X$ is arbitrary. The distribution over $Y$ is unchanged under the following parameter transformations:

$$
\begin{aligned}
A &\leftarrow G^{-1}AG, & C &\leftarrow CG, \\
B &\leftarrow G^{-1}B, & D &\leftarrow D, \\
\mathbf{b} &\leftarrow G^{-1}\mathbf{b}, & \mathbf{d} &\leftarrow \mathbf{d}, \\
R &\leftarrow G^{-1}RG^{-\mathsf{T}}, & S &\leftarrow S.
\end{aligned}
\tag{14}
$$

**Parameterization of $A$.** The stability constraint,

$$\|A\|_2 \le 1, \tag{15}$$

is equivalent to ensuring that the singular values of $A$ lie within the unit hypercube (since singular values are non-negative). Let $A = U\Sigma V^{\mathsf{T}}$ be the singular value decomposition (SVD) of $A$. Now we have from the previous result that if an LDS has latent dynamics with transition parameter $A$, we may replace the dynamics under the similarity transform $G^{-1}AG$. Choose $G = U$, i.e. the left singular values of $A$, and hence $A = \Sigma V^{\mathsf{T}}U =: \Sigma Q$ for some orthogonal matrix $Q$. This follows from the closure of the orthogonal group under multiplication, which is easily verified. Note that in choosing this transformation, no additional constraints are placed on the other parameters in the LDS.

Orthogonal matrices can be parameterized in a number of ways (see e.g. Khuri et al., 1989). A straight-forward choice is the *Cayley transform*. From Khuri et al. (1989): "if $Q$ is an orthogonal matrix that does not have the eigenvalue -1, then it may be written in Cayley's form:

$$Q = (I - \mathcal{S})(I + \mathcal{S})^{-1}, \tag{16}$$

where $\mathcal{S}$ is skew-symmetric". In order to permit negative eigenvalues, we can pre-multiply by a diagonal matrix $E$ with elements in $\{+1, -1\}$. Since we then have $A = \Sigma EQ$, $E$ can be absorbed into $\Sigma$, and so the stability constraint (15) can be satisfied with the parameterization $A = \Sigma Q$ where $\Sigma$ is a diagonal matrix with elements in $[-1, +1]$ and $Q$ is a Cayley-transform of a skew-symmetric matrix. This follows from the overparameterization of the LDS, and we emphasise that the system equations (4) and (6) are not equivalent, but any LDS distribution over $Y$ can be written with latent dynamics of the form (6).

**Parameterization of $B$.** Choose $G = \kappa^{-1}I$ in eq. (14). It may be observed that the scale $\kappa$ of the latent system can be chosen arbitrarily without affecting $A$. We wish to avoid such degeneracies in a hierachical model, since we may otherwise waste statistical strength and computation on learning equivalent representations. We can remove this by fixing the scale of $B$. An indirect but straight-forward approach is to upper bound the magnitude of each element of $B$. For $\tilde{B}$ predicted by $\mathbf{h}_\phi(\mathbf{z})$ we might choose the transformation $B = \tanh(\tilde{B})$ where $\tanh$ acts element-wise. If a sparse $B$ is desired, one can use an over-parameterization of two matrices $\tilde{B}_1, \tilde{B}_2$, and choose $B = \sigma(\tilde{B}_1) \circ \tanh(\tilde{B}_2)$, where $\circ$ is element-wise multiplication, and $\sigma$ a logistic sigmoid. The former parameterization is unlikely to find a sparse representation since the gradient of $\tanh$ is greatest at 0.

**Parameterization of $R$, $S$.** The covariance matrices $R$, $S$ must be in the positive definite cone. Where a diagonal covariance will suffice, any parameterization for enforcing positivity can be used, such as exponentiation, squaring or softplus. A number of parameterizations are available for full covariance matrices (see Pinheiro & Bates, 1996). A simple choice is to decompose the matrix, say $R = LL^{\mathsf{T}}$, where $L$ is a lower triangular Cholseky factor. As before, it is useful to enforce uniqueness, which can be done by ensuring the diagonal is positive.

### A.1.3 LEARNING VIA A MONTE CARLO OBJECTIVE

We provide an alternative learning algorithm to the VB approach in section 2.2 which obtains a tighter lower bound. This was important for the DHO experiments in order to monitor convergence to the true model. The below is perhaps a novel approach for learning in unsupervised cases (i.e. where $U = \varnothing$), but cannot be performed efficiently for supervised problems without modification.

Monte Carlo Objectives (MCOs, Mnih & Rezende, 2016) construct a lower bound for marginal likelihoods via a transformation of an appropriate Monte Carlo estimator. Specifically we consider the logarithmic transformation of:

$$p(Y) \approx \frac{1}{M}\sum_{m=1}^{M}\frac{p(Y, \mathbf{z}_m)}{p(\mathbf{z}_m)} = \frac{1}{M}\sum_{m=1}^{M}p(Y \mid \mathbf{z}_m) \qquad \text{for} \quad \mathbf{z}_m \sim p(\mathbf{z}), \tag{17}$$

$m = 1, \ldots, M$; an importance sampling estimator for $p(Y)$. Using Jensen's inequality, we show that the following is a lower bound on the log marginal likelihood:

$$\mathcal{L}_{\text{MCO}} := \mathbb{E}_{p(\mathbf{z}_{1:M})} \left[ \log \frac{1}{M} \sum_{m=1}^{M} p(Y \mid \mathbf{z}_m) \right] \leq \log \mathbb{E}_{p(\mathbf{z}_{1:M})} \left[ \frac{1}{M} \sum_{m=1}^{M} p(Y \mid \mathbf{z}_m) \right] = \log p(Y) \tag{18}$$

where $p(\mathbf{z}_{1:M}) := p(\mathbf{z}_1)...p(\mathbf{z}_M)$. The tightness of the bound can be increased by increasing the number of samples $M$ (Burda et al., 2016). Assuming $p(\mathbf{z})$ has been re-parameterized (Kingma & Welling, 2014) to be parameter-free, we can easily calculate the gradient (if not, see Mnih & Rezende, 2016). By exchanging integration and differentiation, we can calculate the gradient as:

$$\nabla_{\phi} \mathcal{L}_{\text{MCO}} = \int \cdots \int_{\mathcal{Z}} \nabla_{\phi} \log \left[ \frac{1}{M} \sum_{m=1}^{M} p(Y \mid \mathbf{z}_m) \right] p(\mathbf{z}_1)...p(\mathbf{z}_M) \, \mathrm{d}\mathbf{z}_{1:M} \tag{19}$$

$$= \mathbb{E}_{p(\mathbf{z}_{1:M})} \left[ \frac{\nabla_{\phi} \sum_{m=1}^{M} p(Y \mid \mathbf{z}_m)}{\sum_{m=1}^{M} p(Y \mid \mathbf{z}_m)} \right] \tag{20}$$

$$= \mathbb{E}_{p(\mathbf{z}_{1:M})} \left[ \frac{\sum_{m=1}^{M} p(Y \mid \mathbf{z}_m) \nabla_{\phi} \log p(Y \mid \mathbf{z}_m)}{\sum_{m'=1}^{M} p(Y \mid \mathbf{z}_{m'})} \right] \tag{21}$$

$$= \mathbb{E}_{p(\mathbf{z}_{1:M})} \left[ \sum_{m=1}^{M} \tilde{w}_m \nabla_{\phi} \log p(Y \mid \mathbf{z}_m) \right], \tag{22}$$

where $\tilde{w}_m = p(Y \mid \mathbf{z}_m) / \sum_{m'=1}^{M} p(Y \mid \mathbf{z}_{m'})$. Note that eq. (22) is an importance sampled version of the Fisher identity.

We might expect this estimator to suffer from high variance, since the prior is a poor proposal for the posterior. However, the prior should not be a poor proposal for the *aggregate* posterior, i.e. $\frac{1}{N} \sum_{i=1}^{N} p(\mathbf{z} \mid Y^{(i)})$ (see Seth et al., 2017). In fact, importance sampling from the prior may serve as a useful bias in this case, attracting the posterior distributions which have a large $D_{\text{KL}}(p(\mathbf{z} \mid Y^{(i)}) || p(\mathbf{z}))$ towards the prior.

Our observation is that sampling from the prior can be amortized over each sequence $Y^{(i)}$, $i = 1, \ldots, N$. Specifically, for each particle $\mathbf{z}_m$, the dynamics (2), (3) can be run forward once to calculate $\hat{Y}_m$, from which the likelihood $Y^{(i)}$, *for all tasks* $i = 1, \ldots, N$ can be calculated inexpensively. The amortized cost of taking $M$ samples (e.g. $M \in \mathcal{O}(10^3)$) now becomes $M/N$, which may be relatively small. We can also take advantage of low-discrepancy random variates such as Sobol sequences (Lemieux, 2009) to reduce variance. We propose that each sequence $i$ resamples a small number $M_{\text{rsmp}} \leq 5$ of particles from the importance weights for each $i$ to reduce the cost of backpropagation (a similar resampling scheme is suggested in Burda et al., 2016). See Algorithm 1.

In the supervised case (i.e. where each observation $Y^{(i)}$ has a different input $U^{(i)}$), running the dynamics forward from a particle $\mathbf{z}_m$ can no longer be amortized over all $\{Y^{(i)}\}$ since the prediction $\hat{Y}^{(i)}$ depends on $U^{(i)}$. We can therefore only amortize the parameter generation $\boldsymbol{\theta} = \mathbf{h}_{\phi}(\mathbf{z})$, which is often less expensive than running the dynamics forward. For this reason Algorithm 1 is primarily restricted to unsupervised problems. A hybrid approach would essentially result in the importance weighted autoencoder (IWAE) of Burda et al. (2016).

### A.1.4 INFERENCE

Inference at test time can be performed by any number of variational or Monte Carlo approaches. As in the main text, our focus here is on deterministic state dynamical systems. For stochastic state models, additional reasoning similar to Miladinović et al. (2019) will be required.

A gold standard of inference over $\mathbf{z}$ may be the No U-Turn Sampler (NUTS) of Hoffman & Gelman (2014) (a form of Hamiltonian Monte Carlo), provided $k$ is not too large and efficiency is not a concern. However, given the sequential nature of the model, it is natural to consider exploiting the posterior at time $t$ for calculating the posterior at time $t + 1$. Bayes' rule suggests an update of the

---

**Algorithm 1:** Importance Sampled Optimization for Unsupervised MTDS

---

**Result:** Optimized parameter $\hat{\phi}$
**Inputs**: $\{Y^{(i)}\}_{i=1}^{N}, \phi, M, M_{\text{rsmp}}$, nepochs, optimizer;
**for** *epoch = 1:nepochs* **do**
    **for** *minibatch $S$ in $\{1, \ldots, N\}$* **do**
        // calculate posterior samples;
        $\mathbf{z}_m \overset{\text{Sobol}}{\sim} p(\mathbf{z}), \; m = 1, \ldots, M$;
        $W \leftarrow construct\_weights\left(\log p(Y = \cdot \mid \mathbf{h}_\phi(\cdot)), \{\mathbf{z}_m\}_{m=1}^{M}, \{Y^{(i)}\}_{i \in S}\right)$;
        // compute gradient;
        $\mathbf{g} \leftarrow \mathbf{0}$;
        **for** *$i$ in $S$* **do**
            **for** *$m$ in $\{1, \ldots, M_{rsmp}\}$* **do**
                $i \sim Categorical\left(W_{(i)}\right)$;
                $\mathbf{g} \mathrel{+}= \frac{1}{M_{\text{rsmp}}} \nabla_\phi \log p(Y^{(i)} \mid \mathbf{h}_\phi(\mathbf{z}_i))$;
            **end**
        **end**
        Optimize(optimizer, $\phi$, $\mathbf{g}$);
    **end**
**end**

---

following form:

$$p(\mathbf{z} \mid \mathbf{y}'_{1:t+1}, \mathbf{u}'_{1:t+1}) \; \propto \; p(\mathbf{y}'_{t+1} \mid \mathbf{u}'_{1:t+1}, \mathbf{h}_\phi(\mathbf{z})) \, p(\mathbf{z} \mid \mathbf{y}'_{1:t}, \mathbf{u}'_{1:t}), \tag{23}$$

following the conditional independence assumptions of the MTDS. This update (in principle) incorporates the information learned at time $t$ in an optimal way, and further suggests a constant time update wrt $t$. However, evaluation of $p(\mathbf{y}'_{t+1} \mid \mathbf{u}'_{1:t+1}, \mathbf{h}_\phi(\mathbf{z}))$ usually scales linearly with $t$, since the state $\mathbf{x}_{t+1}$ must be calculated recursively from $\mathbf{x}_0$ given $\mathbf{z}$ and $\mathbf{u}'_{1:t+1}$. Nevertheless, sequential incorporation of previous information will perform a kind of annealing (Chopin, 2002) which reduces the difficulty, and hopefully the runtime of inference at each stage.

We first provide some background of the difficulties of such an approach, looking first at Monte Carlo (MC) methods. Naïve application of Sequential Monte Carlo (SMC) will result in severe particle depletion over time. To see this, let the posterior after time $t$ be $p(\mathbf{z} \mid \mathbf{y}'_{1:t}, \mathbf{u}'_{1:t}) = \frac{1}{M} \sum_{m=1}^{M} w_m \delta(\mathbf{z} - \mathbf{z}_m)$. Then the updated posterior at time $t + 1$ will be:

$$p(\mathbf{z} \mid \mathbf{y}'_{1:t+1}, \mathbf{u}'_{1:t+1}) \; \propto \; \frac{1}{M} \sum_{m=1}^{M} w_m p(\mathbf{y}'_{t+1} \mid \mathbf{u}'_{t+1}, \mathbf{h}_\phi(\mathbf{z})) \delta(\mathbf{z} - \mathbf{z}_m), \tag{24}$$

$$\Rightarrow p(\mathbf{z} \mid \mathbf{y}'_{1:t+1}, \mathbf{u}'_{1:t+1}) \; = \; \frac{1}{M} \sum_{m=1}^{M} \tilde{w}_m \delta(\mathbf{z} - \mathbf{z}_m), \tag{25}$$

where $\tilde{w}_m = \frac{w_m p(\mathbf{y}'_{t+1} \mid \mathbf{u}'_{t+1}, \mathbf{h}_\phi(\mathbf{z}_m))}{\sum_{j=1}^{M} w_j p(\mathbf{y}'_{t+1} \mid \mathbf{u}'_{t+1}, \mathbf{h}_\phi(\mathbf{z}_j))}$, simply a re-weighting of existing particles. Over time, the number of particles with significant weights $w_m$ will substantially reduce. But since the model is static with respect to $\mathbf{z}$ (see Chopin, 2002), there is no dynamic process to 'jitter' the $\{\mathbf{z}_m\}$ as in a typical particle filter, and hence a resampling step cannot improve diversity.

Chopin (2002) discusses two related solutions: firstly using 'rejuvenation steps' (cf. Gilks & Berzuini, 2001) which applies a Markov transition kernel to each particle. The downside to this approach is the requirement to run until convergence; and the diagnosis thereof, which can result in substantial extra computation. One might instead sample from a fixed proposal distribution (accepting a move with the usual Metropolis-Hastings probability) for which convergence is more easily monitored. A Sequential Monte Carlo sampler approach (Del Moral et al., 2006) may be preferred, which permits local moves, and can reduce sample impoverishment via resampling (similar to SMC). However, the approach requires careful choices of both forward and backward Markov kernels which substantially reduces its ease of use.

A well-known variational approach to problems with the structure of eq. (23) is assumed density filtering (ADF, see e.g. Opper & Winther, 1998). For each $t$, ADF performs the Bayesian update and the projects the posterior into a parametric family $\mathcal{Q}$. The projection is done with respect to the *reverse* KL Divergence, i.e. $q_{t+1} = \arg\min_{q \in \mathcal{Q}} D_{\mathrm{KL}}\left(p(\mathbf{z} \,|\, \mathbf{y}'_{1:t+1}, \mathbf{u}'_{1:t+1}) \,\|\, q\right)$. Intuitively, the projection finds an 'outer approximation' of the true posterior, avoiding the 'mode seeking' behaviour of the forward KL, which is particularly problematic if it attaches to the wrong mode. Clearly the performance of ADF depends crucially on the choice of $\mathcal{Q}$. Unfortunately, where $\mathcal{Q}$ is expressive enough to capture a good approximation, the optimization problem will usually be challenging, and must resort to stochastic gradient approaches, resulting in an expensive inner loop. Furthermore, when the changes from $q_t$ to $q_{t+1}$ are relatively small, the gradient signal will be weak, resulting perhaps in misdiagnosed convergence and hence accumulation of error over increasing $t$. A recent suggestion of Tomasetti et al. (2019) is to improve efficiency via re-use of previous (stale) gradient evaluations. Standard variance reduction techniques may also be considered to improve convergence in the inner loop.

### A.1.5 ONLINE INFERENCE – OUR APPROACH

In our experiments, we found sampling approaches faster and more reliable for each update, as well as providing diagnostic information, and so we eschew variational approaches. (Our experiments used a fairly small $k$ ($\leq 10$); variational approaches may be preferred in higher dimensional problems.) Specifically we use iterated importance sampling (IS) to update the posterior at each $t$. The key quantity for IS is the proposal distribution $q_{\mathrm{prop}}$: we need a proposal that is well-matched to the target distribution. Our observation is that the natural annealing properties of the filtering distributions (eq. 23) allow a slow and reliable adaptation of $q_{\mathrm{prop}}$.

In order to capture complex multimodal posteriors, we parameterize $q_{\mathrm{prop}}$ by a mixture of Gaussians (MoG). For each $t$, the proposal distribution is improved over $N_{\mathrm{AIS}}$ iterations using adaptive importance sampling (AdaIS), described for mixture models in Cappé et al. (2008). We briefly review the methodology for a target distribution $p_*$. Let the AdaIS procedure at the $n$th iteration use the proposal:

$$q_{\mathrm{prop}}^n(\mathbf{z}) \;:=\; \sum_{j=1}^{J} \alpha_j^n \mathcal{N}\left(\boldsymbol{\mu}_j^n, \Sigma_j^n\right), \tag{26}$$

$\alpha_j \in \mathbb{R}_+$ s.t. $\sum_{j=1}^{J} \alpha_j = 1$. For iteration $n$, sample $\mathbf{z}_m \sim q_{\mathrm{prop}}^{n-1}$, $m = 1, \ldots, M$, and calculate the (self-normalized) importance weights $\tilde{w}_m \propto p_*(\mathbf{z}_m)/q_{\mathrm{prop}}^{n-1}(\mathbf{z}_m)$. The resulting empirical distribution is then used to fit $q_{\mathrm{prop}}^{n+1}$, estimating $\{\alpha_j^{n+1}, \boldsymbol{\mu}_j^{n+1}, \Sigma_j^{n+1}\}_{j=1}^{J}$ via (weighted) Expectation Maximization (EM, see Cappé et al., 2008, for details). We monitor the effective sample size (ESS, see ch. 9, Owen, 2013) every iteration and stop once the ESS has reached a certain threshold $M_{\mathrm{ess}}$, see Algorithm 2.

For our experiments, this approach worked robustly and efficiently, and appears superior to the alternatives discussed. Unlike SMC, we obtain a $q_{\mathrm{prop}}$ which is a good *parameteric* approximation of the true posterior. We therefore avoid the sample impoverishment problem discussed above (eq. 25). Due to the small number of iterations of AdaIS required (usually $\leq 5$ for our problems), it is substantially faster than MCMC moves, and since stochastic gradients are avoided, convergence is much faster than variational approaches. The scheme benefits from the observed fast initial convergence rates of the EM algorithm (see e.g. Xu & Jordan, 1996), particularly since early stopping can be used for the initial iterates.

In practice, one may not wish to calculate a posterior at every $t$, but instead intervals of length $\tau$. In our DHO experiments ($k = 4$) we use $\tau = 5$, and usually have ESS $> 0.6M$ after $n = 4$ inner iterations, with total computation per $q_t$ requiring 250-300ms on a laptop. We observe in our experiments that posteriors are often multimodal for $t \leq 20$ and sometimes beyond, motivating the MoG parameterization. In these experiments, the MoG appears to capture the salient characteristics of the target distribution well. Note as in section A.1.3, Sobol or other low-discrepancy sequences may be used to reduce sampling variance from $q_{\mathrm{prop}}$.

---

**Algorithm 2:** Filtered inference via Iterated AdaIS.

---

**Result:** Approximate posteriors $\{q_t\}_{t=1}^T$
**Inputs**: $\mathbf{y}_{1:T}$, $\mathbf{u}_{1:T}$, $\phi$, $M$, $M_{\text{ess}}$, $N_{\text{AIS}}$, $J$;
$q_0 \leftarrow p(\mathbf{z})$;
**for** $t = 1 : T$ **do**
    ess $\leftarrow 0$;
    $q_{\text{prop}}^0 \leftarrow q_{t-1}$;
    **for** $n = 1 : N_{AIS}$ **do**
        **for** *m = 1:M* **do**
            $\mathbf{z}_m \sim q_{\text{prop}}^{n-1}$;
            $w_m \leftarrow \frac{p(\mathbf{y}_{1:t} \mid \mathbf{u}_{1:t}, \mathbf{h}_\phi(\mathbf{z}_m))p(\mathbf{z})}{q_{\text{prop}}^{n-1}(\mathbf{z}_m)}$;
        **end**
        $\tilde{w}_m \leftarrow \frac{w_m}{\sum_{\ell=1}^M w_\ell}$, $\quad m = 1, \ldots, M$;
        $q_{\text{prop}}^n \leftarrow \text{WeightedExpectationMaximization}\left(\{\mathbf{z}_m\}_{m=1}^M, \{\tilde{w}_m\}_{m=1}^M, J; \text{ init} = q_{\text{prop}}^{n-1}\right)$;
        ess $\leftarrow \text{EffectiveSampleSize}\left(\{\tilde{w}_m\}_{m=1}^M\right)$;
        **if** *ess > $M_{ess}$* **then**
            break;
        **end**
    **end**
    $q_t \leftarrow q_{\text{prop}}^n$
**end**

---

## A.2 DAMPED HARMONIC OSCILLATION

This section provides further details about the damped harmonic oscillator (DHO) experiments: the data, experimental setup and results.

### A.2.1 DATA

We generate data via the model:

$$y_t^{(i)} = (\rho_1^{(i)})^t \sin(\omega_2^{(i)} t) - 0.5(\rho_2^{(i)})^t \sin(\omega_2^{(i)} t) + \epsilon_t^{(i)}, \tag{27}$$

$t = 1, \ldots, 80$ which is the sum of two damped harmonic oscillators each with angular frequency $\omega_j$ and decay factor $\rho_j \in (0, 1]$ for tasks $i = 1, 2, \ldots, N$. The data are corrupted by iid Gaussian noise, $\epsilon_t^{(i)} \sim \mathcal{N}\left(0, 0.05^2\right)$. The distribution over the random variables is given in Figure 5c. We choose the second component to be (in expectation) a higher frequency, faster decaying component. A visualization of these distributions is provided in Figures 5a-5b. The distributions were chosen to achieve good visual diversity of sequences.

It is natural to parameterize $\rho_j$ by its half-life (i.e. $t$ such that $\rho_j^t = 0.5$), since interesting decay factors are concentrated near 1. For instance, $\rho_j = 0.98$ results in a half-life of $\nu \approx 34$, and $\rho_j = 0.99$ results in a half-life of $\nu \approx 69$. To generate a decay factor, we sample a half life $\nu$ in a relevant interval (Table 5c) and, using the definition of half life, transform via $\rho = \exp\{-\log(2)/\nu\}$.[1]

### A.2.2 EXPERIMENTAL SETUP

**Parameterization.** We use the model:

$$\{A, B, C\} = \mathbf{h}_\phi(\mathbf{z}) \tag{28}$$
$$\mathbf{x}_t = A \mathbf{x}_{t-1} + B u_t \tag{29}$$
$$\mathbf{y}_t \sim \mathcal{N}\left(C \mathbf{x}_t, s^2\right) \tag{30}$$

---

[1]The distributions have support $\rho_1 \in [0.8409, 0.9914]$, $\rho_2 \in [0.9170, 0.9885]$ (4 decimal places) with density $p(\rho_1) = \frac{\log(2)}{76} \frac{1}{\rho_1 \log^2(\rho_1)}$ and $p(\rho_2) = \frac{\log(2)}{52} \frac{1}{\rho_1 \log^2(\rho_1)}$.

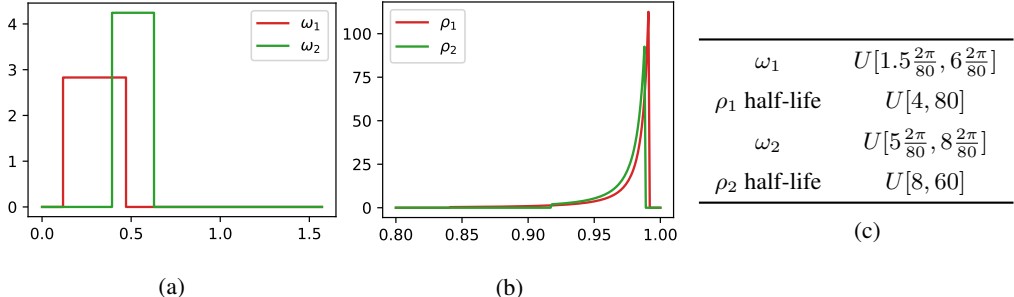

Figure 5: Distribution of random variables for DHO data generation.

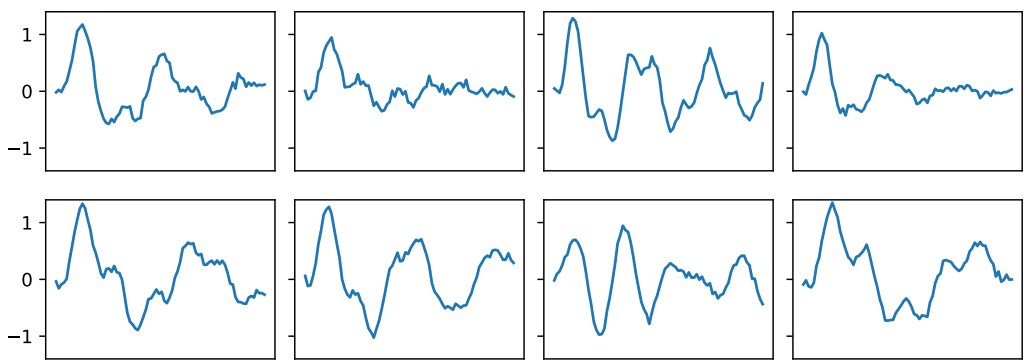

Figure 6: Example data sampled from DHO model.

for $t = 1, \ldots, T$, $\mathbf{z} \in \mathbb{R}^4$, $\mathbf{x}_t \in \mathbb{R}^4$, suppressing task index $i$ for clarity. Define $\mathbf{x}_0 := \mathbf{0}$, and for all tasks $u = [1, 0, 0, 0, \ldots]$. $A$ is parameterized as discussed in section A.1.2 using a product of a diagonal and orthogonal matrix $\Sigma Q$. The diagonal of $\Sigma$ is constrained to lie in $[-1, +1]$ using the $\tanh$ function, and $Q$ is parameterized by the Cayley transform of a skew symmetric matrix $\mathcal{S}$. Using an upper triangular matrix $\Gamma$, we have $\mathcal{S} = \Gamma - \Gamma^\mathsf{T}$, and $Q = (I - \mathcal{S})(I + \mathcal{S})^{-1}$. We parameterize $B$ via the product of logistic sigmoid and $\tanh$ functions as in section A.1.2 in order to learn a sparse parameterization. $C$ is unconstrained, and the parameter $s$ is optimized as a constant wrt. $\mathbf{z}$. The STLDS is parameterized in the same way. The prior $p(\mathbf{z})$ is a unit Gaussian distribution, and $\mathbf{h}_\phi$ is a 2 hidden-layer neural network. We use a fixed feature extractor $\mathbf{z} \rightarrow [\mathbf{z}^\mathsf{T}, \sin(\mathbf{z}^\mathsf{T}), \cos(\mathbf{z}^\mathsf{T}), \|\mathbf{z}\|]^\mathsf{T}$ in the first layer in order to help encode a rectangular support within a spherically symmetric distribution. The second layer is a fully-connected 300 unit layer with sigmoid activations.

**Learning.** The output of an MTLDS is very sensitive to the parameter $A$, and care must be taken to avoid divergence during optimization. The diagonal-orthogonal parameterization greatly helped to stabilize the optimization over a more naïve orthogonal-diagonal-orthogonal SVD parameterization. We also reduced the learning rate by a factor of 10 for $A$. It proved useful to artificially elevate the estimate of $s$ during training using a prior $\log s \sim \mathcal{N}(-1.5, 0.05)$ (derived from preliminary experiments) since the MTDS can otherwise overfit small datasets (see also discussion in §3, Svénsen, 1998), with associated instability in optimization. The learning rate schedule is given in Table 2, for which the prior over $\log s \sim \mathcal{N}(m, 0.05)$ was annealed from $m = -1.0$ to $m = -1.5$. The "momentum" parameter $\beta_1$ (c.f. Kingma & Ba, 2014) is also reduced at the end of optimization. The latter was motivated by oscillation and moderate deviations observed near optima, apparently caused (upon investigation) by strong curvature of the loss surface.

**Inference.** The latent $\mathbf{z}$ are inferred online using the adaptive IS scheme of section A.1.5. We also perform inference over $\log s$ since it is held artificially high for optimization, and its true optimal value is not known. An informative prior close to the learned value $\log s \sim \mathcal{N}(-2.0, 0.1^2)$, was

| Epoch | $\eta$ | $\beta_1$ | $\log s$ mean | $M$ |
|---|---|---|---|---|
| 1 | 8e-4 | 0.9 | -1.0 | 1 000 |
| 200 | 8e-4 | 0.9 | -1.3 | 1 000 |
| 600 | 4e-4 | 0.9 | -1.5 | 2 000 |
| 1000 | 2e-4 | 0.8 | -1.5 | 4 000 |

Table 2: DHO optimization schedule, see text, $\eta$ is learning rate, $M$ is number of samples in MCO step (A.1.3).

| Parameter | Description | Value |
|---|---|---|
| $J$ | Num. mixture components | 3 |
| $N_{AIS}$ | Max. num. of adaptive IS iterations | 7 |
| $M$ | Num. samples per IS iteration | 1 000 |
| $M_0$ | Num. samples for first proposal | 3 000 |
| $M_{\text{final}}$ | Num. samples for final proposal | 3 000 |
| tilt | Exponential tilt of proposal | 2.0 |
| $M_{\text{ess}}$ | Minimum eff. sample size | 100 |
| n_retry | Num. retries if ESS $< M_{\text{ess}}$ | 2 |
| EM_iters | Num. EM iters in GMM fit | 3 |
| kmeans_iters | Max. kmeans iters for init | 100 |

Table 3: DHO Inference parameters.

nevertheless used since the posterior was sometimes approximately singular, causing high condition numbers in the estimated covariance matrix of the proposal. The hyperparameters are given in Table 3. These parameters did not require tuning as for optimization, but were sensible defaults. These also seem to work well without tuning for other experiments such as the Mocap data. Each posterior for a given time $t$ took on average approx. 0.3 seconds.

We used the No U-Turn Sampler (Hoffman & Gelman, 2014) for the STL experiments due to poor conditioning and the higher complexity and dimensionality of the posterior (19 dimensions). Tuning is performed using ideas from Hoffman & Gelman (2014, Algorithm 4, 5), and the mass matrix is estimated from the warmup phase.[2] The warmup stage lasted 1000 samples and the subsequent 600 samples were used for inference. Each sampler was initialized from a MAP value, obtained via optimization with 10 random restarts. For both MAP optimization and sampling, we found it essential to enforce a low standard deviation (we used $\log s = -2$ and $\log s \sim \mathcal{N}\left(-2, 0.2^2\right)$ respectively) similarly to the MTL experiments. The autocorrelation-based effective sample size (Gelman et al., 2013, ch. 11.5) typically exceeds 100 for each parameter. Each posterior for a given time $t$ took on average approx. 300 seconds. Note that as discussed in section A.1.4, unlike our AdaIS procedure, we cannot make much re-use of previous computation here.

### A.2.3 RESULTS

The average results (over the 10 repetitions) are given in Table 4, which extends Table 1 in the main text with the NLL results. The distribution of these results can be seen in the violin plots of Figure 8. The RMSE results of the MTLDS are all significantly better than both the pooled and single-task models according to a Welch's t-test and Mann-Whitney U-test, except for MTLDS-4 at $t = 40$. The latter is significantly better than the pooled model, but is indistinguishable from the STLDS at the level $\alpha = 0.05$.

We also consider the convergence of the MTLDS to the true model with increasing $N$. For each experiment, we average the log marginal likelihood of the test sequences estimated via 10 000 (Sobol) samples from the prior. As before, the prior should be a good proposal for the *aggregate* posterior, and we amortize the same samples over all test sequences. In order to interpret the difference to the true distribution $\log p_*(Y_{\text{test}}) - \log p(Y_{\text{test}} \mid \phi)$, we use the Bayes Factor interpretations given by Kass

---

[2]Implementation `https://github.com/tpapp/DynamicHMC.jl`, author Tamas K. Papp.

| | **RMSE** | | | **NLL** | | |
|---|---|---|---|---|---|---|
| **Model** | 10 | 20 | 40 | 10 | 20 | 40 |
| Pooled-LDS-1k | 0.36 | 0.34 | 0.29 | **0.38** | 0.34 | 0.22 |
| STLDS | 0.43 | 0.36 | 0.11 | 1.98 | 1.24 | -0.37 |
| MTLDS-4 | 0.30 | 0.18 | 0.12 | 1.47 | 0.02 | -0.54 |
| MTLDS-16 | 0.25 | 0.11 | 0.07 | 0.94 | -0.43 | -0.81 |
| MTLDS-128 | **0.23** | **0.09** | **0.06** | 0.83 | **-0.50** | **-0.85** |

Table 4: DHO test results. Predictive RMSE and NLL after $t = 10, 20, 40$. Model suffix denotes training set size.

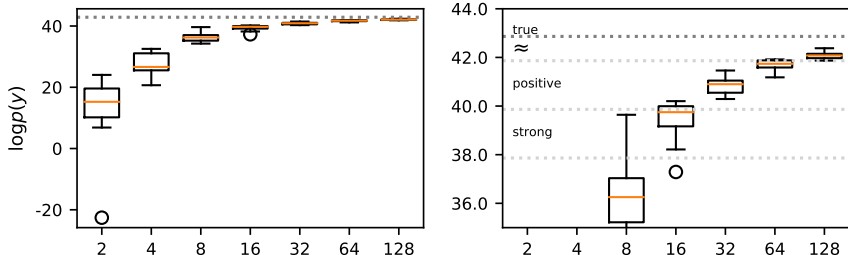

Figure 7: Marginal likelihood of DHO generating distribution under MTLDS models learned from $N$ examples. Boxes show median, IQR and whiskers show most extreme point within $1.5\times$ IQR above/below each box. True value shown as (uppermost) dotted line. The RHS panel is rescaled to show the upper end of the plot with Bayes Factor interpretations overlaid.

& Raftery (1995). For instance a difference of 1.0 is 'barely worth mentioning', but a difference of 4.0 is 'strong evidence' that the distributions are different. We average over 10 000 test examples to avoid sampling variation of the test set. Figure 7 show boxplots of the log marginal likelihood for each model over increasing $N$, where the boxes show the interquartile range (IQR) over the 10 repetitions. We see convergence towards the true value with increasing $N$, with the difference of the MTLDS-128 'barely worth mentioning'.

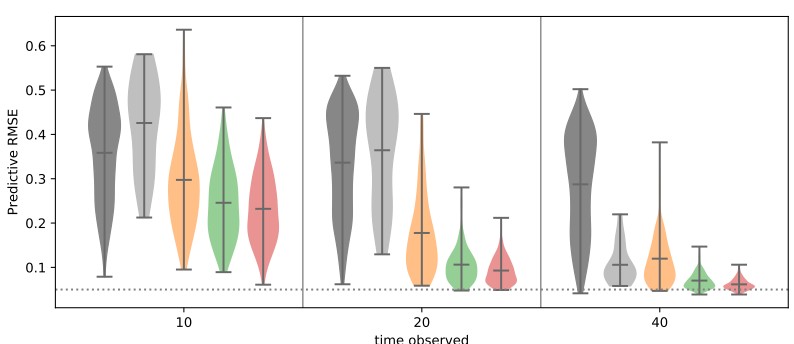

(a) RMSE. Reference (dotted) line shows minimum error for $s = 0.05$.

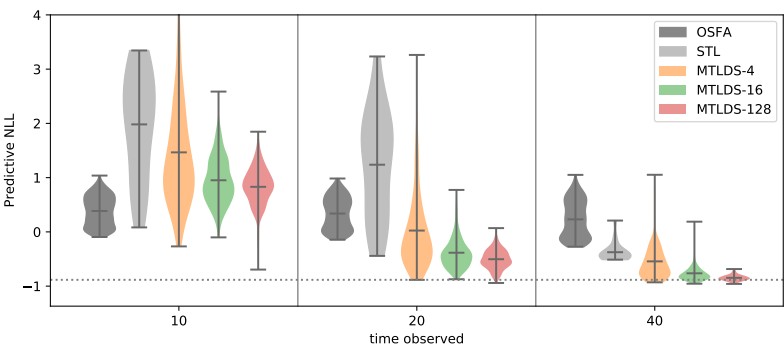

(b) NLL. Reference (dotted) line shows Gaussian entropy with $s = 0.1$.

Figure 8: Improvement in predictive performance over time seen (x-axis) for the DHO experiments. Violin plots show a kernel density estimate of the score achieved on RMSE and NLL over sequences in the test set. Horizontal bars show min, median and max scores.

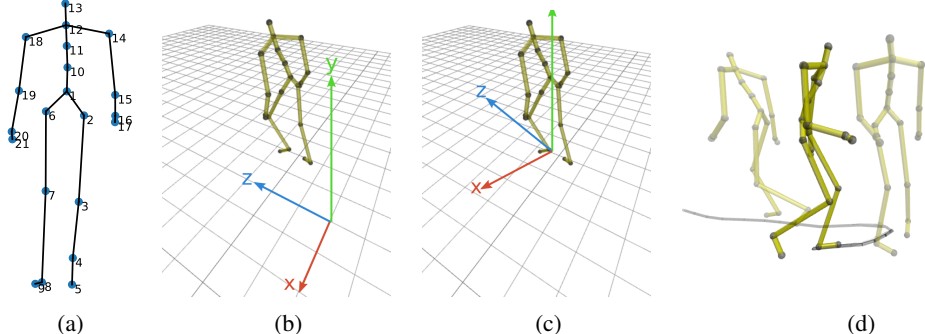

Figure 9: *(a)* The 21-joint skeleton. *(b)* Eulerian representation. *(c)* Lagrangian representation. *(d)* Example of rotating towards the outside of a corner.

### A.3 Human Locomotion Motion Capture (Mocap) Experiments

We provide further details about the data, experimental setup and results of the human locomotion experiments below.

#### A.3.1 Human Locomotion Motion Capture (Mocap)

The mocap data of Mason et al. (2018) consists of planar walking and running motion in 8 styles. The original data is recorded at 120 fps, we downsample to 30 fps as per Martinez et al. (2017); Pavllo et al. (2018). Unlike Mason et al. (2018), we do not perform any data augmentation via mirroring. The data is mapped to a 21-joint skeleton used in the codebase of Holden et al. (2016), shown in Figure 9a, which is a subset of the CMU skeleton.

**Representation in observation space.** We choose a Lagrangian representation (Figure 9c) where the coordinate frame is centered at the *root joint* of the skeleton (joint 1 in Fig. 9a, the pelvis), projected onto the ground. The frame is rotated such that the $z$-axis points in the "forward" direction, roughly normal to the body. This is in contrast to the Eulerian frame (Figure 9b) which has an absolute fixed position for all $t$. In the Lagrangian frame, the joint positions are always relative to the root joint, which avoids confusing the overall *trajectory* of the skeleton (typified by the root joint), and the overall *rotation* of the skeleton, with the local motions of the joints.

The relative joint positions can be represented by spatial position or by joint angle. For the latter, the spatial positions of all joints can be recovered from the angle made with their parent joint via use of forward kinematics (FK). This construction ensures the constant bone length of the skeleton over time, which is a desirable property. However, it also substantially increases the sensitivity of internal joints. For instance, the rotation of the trunk will disproportionately affect the error of the joints in both arms. For this reason, we have chosen to model the spatial position of joints, which may result in violations of bone length, but avoids these sensitivity issues. See also §2.1 Pavllo et al. (2018).

One can further encode the joint positions via velocity (i.e. differencing) which may result in smoother predictions. We avoid this encoding for the local joint motion (joints 2 to 21) since it can suffer from accumulated errors, but we do use it to predict the co-ordinate frame as is standard in mocap models. Hence our per-frame representation consists of the velocity $\dot{x}, \dot{z}, \dot{\omega}$ of the co-ordinate frame, the relative vertical position of the root joint, and 3-d position of the remaining 20 joints, which gives $\mathbf{y}_t \in R^{64}$.

**Choice of inputs.** Our choice of inputs will reflect controls that an animator may wish to manipulate. The first input will be the trajectory that the skeleton is to follow. As in Holden et al. (2017b), we provide the trajectory over the next second (30 frames), sampled uniformly every 5 frames. Unlike previous work, there is no trajectory history in the inputs since this can be kept in the recurrent state. The (2-d) trajectory co-ordinates are given wrt. the current co-ordinate frame, and hence can rotate rapidly during a tight corner. In order to provide some continuity in the inputs, we also provide a first difference of the trajectory in Eulerian co-ordinates.

| Model | Optimizer | $\eta$ | Multi-task $\eta$ | Regularization |
|---|---|---|---|---|
| MT-RNN | Adam | 3e-5 | 1e-3 | 1e-2 |
| GRU L1 (closed loop) | Adam | 5e-4 | - | 5e-4 |
| GRU L2 (closed loop) | Adam | 1e-4 | - | 5e-4 |
| GRU L1 (open loop) | Adam | 5e-4 | - | 0 |
| GRU L2 (open loop) | Adam | 1e-4 | - | 0 |

Table 5: Hyper-parameters of mocap models. $\eta$ denotes the learning rate.

The velocity implied by the differenced trajectory does not disambiguate the gait frequency vs. stride length. The same motion might be achieved with fast short steps, or slower long strides. We therefore provide the gait frequency via a phasor (as in Holden et al., 2017b), whose frequency may be externally controlled. This is provided by sine and cosine components to avoid the discontinuity at $2\pi$. A final ambiguity exists from the trajectory at tight corners: the skeleton can rotate either towards the focus of the corner, or towards the outside. Figure 9d demonstrates the latter, which appears not infrequently in the data. We provide a boolean indicator alongside the trajectory which identifies corners for which this happens. Altogether we have $\mathbf{u}_t \in \mathbb{R}^{32}$: 12 inputs for the Lagrangian trajectory, 12 inputs for the differenced Eulerian trajectory, 2 inputs for the gait phase and 6 inputs for the turning indicators.

**Extracting the root trajectory.** The root trajectory is computed by projecting the root joint onto the ground. However, this projection may still contain information about the style of locomotion, for instance via swaying. We wish to remove all such information, since a model can otherwise learn the style without reference to a latent $\mathbf{z}$. Our goal is to find an *appropriately smoothed* version of the extracted trajectory $\mathcal{T}$. We use a cubic B-spline fit to control points fitted to the 'corners' of the trajectory. These control points are selected using a polygonal approximation to $\mathcal{T}$ using the Ramer-Douglas-Peucker algorithm (RDP, e.g. Ramer, 1972). Briefly, the RDP algorithm uses a divide-and-conquer approach which greedily chooses points that minimize the Hausdorff distance of $\mathcal{T}$ to the polygonal approximation. Some per-style tuning of the RDP parameter, and a small number of manually added control points rendered this a semi-automatic process.

**Extracting the gait phase.** Foot contacts are calculated via the code used by Holden et al. (2017b), which is based on thresholding of vertical position and velocity of each foot. As in Mason et al. (2018) we check visually for outliers and correct misclassified foot contacts manually. The leading edge of each foot contact is taken to represent 0 (left) and $\pi$ (right), and the gait phase is calculated by interpolation.

### A.3.2 EXPERIMENTAL SETUP

In this section we discuss elements of the experimental setup, learning and inference common to all experiments. Details particular to each experiment can be found in the following section.

**Further Model Details** The MTDS architecture is described in section A.3, aside from the choice of prior. We tested both linear and nonlinear $\mathbf{h}_\phi$ in preliminary experiments and the performance was often similar. The nonlinear version used a one hidden layer MLP with 300 hidden units with $\tanh$ activations. For the final affine layer, we used a rank 30 matrix which, chosen pragmatically as a trade-off between flexibility and parameter count (see discussion in section A.1.1). Both choices often performed similarly, however the linear approach was chosen, since optimization of the latent $\mathbf{z}$ on new data was faster, and apparently more robust to choice of initialization. A nonlinear $\mathbf{h}_\phi$ may be more important when the base model is simpler.

The benchmark models use an encoding length of $\tau = 64$ frames. The encoder shares parameters with the decoder, i.e. the RNN is simply 'warm started' for 64 frames before prediction. The benchmark models, unlike the MTDS, predict the difference from the previous frame (or 'velocity') via a residual architecture, as this performs better in Martinez et al. (2017).

**Further Learning Details**   Our primary goal was qualitative: to obtain good style-content separation, high quality animations and smooth interpolation between sequences. Therefore hyperparameter selection for the MTDS proceeded via quantitative means (via the ELBO) and visual inspection of the qualitative criteria. The qualitative desiderata motivated split learning rates between shared and multi-task networks (cf. section 4.2), and the amount of L2 regularization. See Table 5 for the chosen values. The main learning rate $\eta$ applies to the fixed parameters wrt. $\mathbf{z}$ (i.e. $\psi_1, H$), and the multi-task learning rate applies to the parameter generation parameters $\phi$ and inference parameters $\lambda$. Standard variational inference proved more reliable than amortized inference: we used a Gaussian with diagonal covariance (parameterized using softplus) for the variational posterior over each $\mathbf{z}$. L2 regularization was applied to $\phi, \psi_1, H$.

Unless otherwise specified, we optimized each model using a batch size $N_{\text{batch}} = 16$ for $20\,000$ iterations. The ELBO had often reached a plateau by this time, and training even longer resulted in a worse latent representation at times (as evidenced through poor style transfer). As noted in the main text, we remove the KL penalty of eq. (9) for the initial $2\,000$ iterations, and enforce a small posterior standard deviation ($\mathbf{s}_\lambda = 10^{-3}$) for the same duration. This is similar to finding a MAP estimate for the $\{\mathbf{z}\}$. For the remaining iterations, the original ELBO criterion is used, and the constraint on $\mathbf{s}_\lambda$ is removed. The model is implemented in PyTorch (Paszke et al., 2017) and trained on GPUs. Since we use a fairly small max. sequence length $L = 64$, truncated backpropagation through time was not necessary.

The hyper-parameters for the benchmark models were found (Table 5) using a grid search over learning rate and regularization, as well as the optimizers {Adam, (vanilla) SGD}. We performed the search over the pooled data for all 8 styles, with a stratified sample of 12.5% held out for a validation set. Once the hyperparameters were chosen, benchmark models were also trained for $20\,000$ iterations, recording the validation error every $1\,000$ iterations on a stratified 12.5% held out sample. The model with the lowest validation error during optimization is chosen.

We standardize the data so that when pooled, each dimension has zero mean and unit variance. Finally, note that as discussed in section A.3.1, the data are represented in Lagrangian form, therefore drifts in the predicted trajectory from the true one are not necessarily heavily penalized. This can be altered by changing the weights on the root velocities, but we did not do this.

**Inference.**   At test time, especially for experiment 2, we cannot expect amortized inference to perform optimally, and we consider standard inference techniques. We want to understand the nature of the posterior distributions, and so we again used the AdaIS approach of section A.1.5. In practice, each posterior was unimodal and approximately Gaussian. Furthermore, the variation in sequence space for different $\mathbf{z}$ in the posterior was usually fairly small, and the posterior predictive mean performed similarly to using a point estimate. Each observation from which $\mathbf{z}$ is inferred is of size $64 \times 64$ and hence the posterior is fairly concentrated. Unlike the DHO model, this is a more expensive procedure. Our $k = 3$ experiments took approx. 24 seconds per observation for inference. An optimization approach using standard techniques may be expected to perform similarly at a reduced computational cost. Hence unless otherwise specified, inference was done via optimization.

## A.4   EXPERIMENTS

**Experiment 1 – MTL**   The training data for each style uses 4 subsequences chosen carefully to represent the inter-style variation. Obviously it is important that frames are consecutive rather than randomly sampled. Over the increasing size training sets, each of these subsequences is a superset of the previous one. The 6 training set sizes ($2^8, 2^9, 2^{10}, 2^{11}, 2^{12}, 2^{13}$ frames per style) are not exact since short subsequences are discarded (e.g. at file boundaries), and the largest set contains all the training data except the test set[3], where data are not evenly distributed over styles. The test set comprises 4 sequences from each style, each of length 64, and is the same for all experiments. A length-64 seed sequence immediately preceding each test sequence was used for inference for all models. The models are trained as described above, except for the single task (STL) models. The STL models use an identical architecture to the pooled 1-layer GRU models, except they are trained only on the data for their style. Since there is less data for these models, we train them for a

---

[3]This final set averages $7680 \approx 2^{12.9}$ frames per style.

| Model | RMSE | | | | | |
|---|---|---|---|---|---|---|
| | Training set size | | | | | |
| | 3% | 7% | 13% | 27% | 53% | 97% |
| Training mean | 0.76 | 0.76 | 0.72 | 0.73 | 0.73 | 0.73 |
| Zero-velocity | 1.23 | 1.23 | 1.23 | 1.23 | 1.23 | 1.23 |
| Pooled GRU (closed loop) | 0.79 | 0.61 | 0.82 | 0.87 | 0.76 | 1.21 |
| STL GRU (open loop) | 1.11 | 0.88 | 0.40 | 0.33 | **0.18** | **0.18** |
| Pooled GRU (open loop) | 0.69 | 0.52 | 0.36 | 0.29 | **0.16** | **0.16** |
| MT Bias ($k=3$) | 0.93 | 0.44 | 0.30 | **0.21** | **0.14** | **0.16** |
| MT Bias ($k=5$) | 0.98 | 0.44 | 0.30 | **0.20** | **0.14** | **0.16** |
| MT Bias ($k=7$) | 0.94 | 0.49 | 0.30 | **0.21** | **0.15** | **0.16** |
| MTDS ($k=3$) | 0.62 | 0.34 | 0.35 | **0.21** | 0.21 | 0.19 |
| MTDS ($k=5$) | **0.53** | **0.29** | **0.22** | **0.19** | **0.15** | **0.16** |
| MTDS ($k=7$) | **0.51** | **0.27** | **0.24** | **0.20** | **0.16** | **0.18** |

Table 6: Mocap Experiment 1 (MTL): predictive MSE for length-64 predictions where training sets are a given fraction of the original dataset.

maximum of 5 000 iterations. We do not train 2-layer GRUs, since the amount of data is small for most experiments.

The full results are given in Table 6. We use fractions of the dataset instead of absolute training set sizes to aid understanding. The performance of the MTDS appears to increase with larger $k$, and suggests that we need $k > 3$ to achieve optimal performance on unseen training data. The results demonstrate substantial benefit of the MTDS over a pooled RNN model in terms of sample efficiency, but not in asymptotic performance, as might be expected. According to a paired t-test, the improvements of the $k = 7$ MTDS over the (1-layer, open loop) pooled GRU are significant for training set sizes 3%, 7%, 13% and 27%.[4] At a style level, the $k = 7$ MTDS performs at least as well as the pooled GRUs for the first four training set sizes. See Figure 10. Note that the 'angry' and 'childlike' styles appear to be harder than the others, most likely due to their relatively high speed. For example animations of the MTL experiments, see the linked video in section A.4.1.

**Experiment 2 – Novel Sequences**    Table 7 provides the aggregate results of experiment 2 for each of the mocap models. A visualization is given in Figure 11. The 2-layer competitors are shown here for completeness, but they achieve similar performance to the 1-layer models on aggregate. Figure 12 provides a breakdown of these results on a per-style basis. Styles 5-8 appear to be easier from the point of view of the benchmarks, but the MTDS shows equal or better performance on all styles except style 5.

The competitor results achieve better short-term performance than the MTDS. However, note that the zero-velocity baseline performs similarly to the open-loop GRUs for the first 5 predictions. This suggests that the MTDS may be improved for these early predictions simply by interpolating from the zero-velocity baseline for small values of $t$. We are unable to conclude from these experiments that the benchmark models can represent the style better initially, but simply that they can smooth the transition from the seed sequence better.

**Experiment 3 – Style Transfer**    The classifier is learned on the original observations to distinguish between the 8 styles. We use a 512-unit GRU to encode an observation sequence (usually of 64 frames), and transform the final state via a 300-unit hidden layer MLP with sigmoid activations into multinomial emissions. The model is trained via cross-entropy with 20% of the training data held out as a validation set; training was stopped as the validation error approached 0. We perform a standardization of the gait frequency across all styles, since some styles can be identified purely by calculating the frequency. The mean frequency across all styles (1 cycle per 33 frames) is applied to all sequences via linear interpolation. In this we make use of the instantaneous phase given in the inputs. We use a $k = 8$ latent code for the MTDS as the model is trained on all styles.

---

[4]A non-parameteric Mann-Whitney U test gives 7%, 13% as significant.

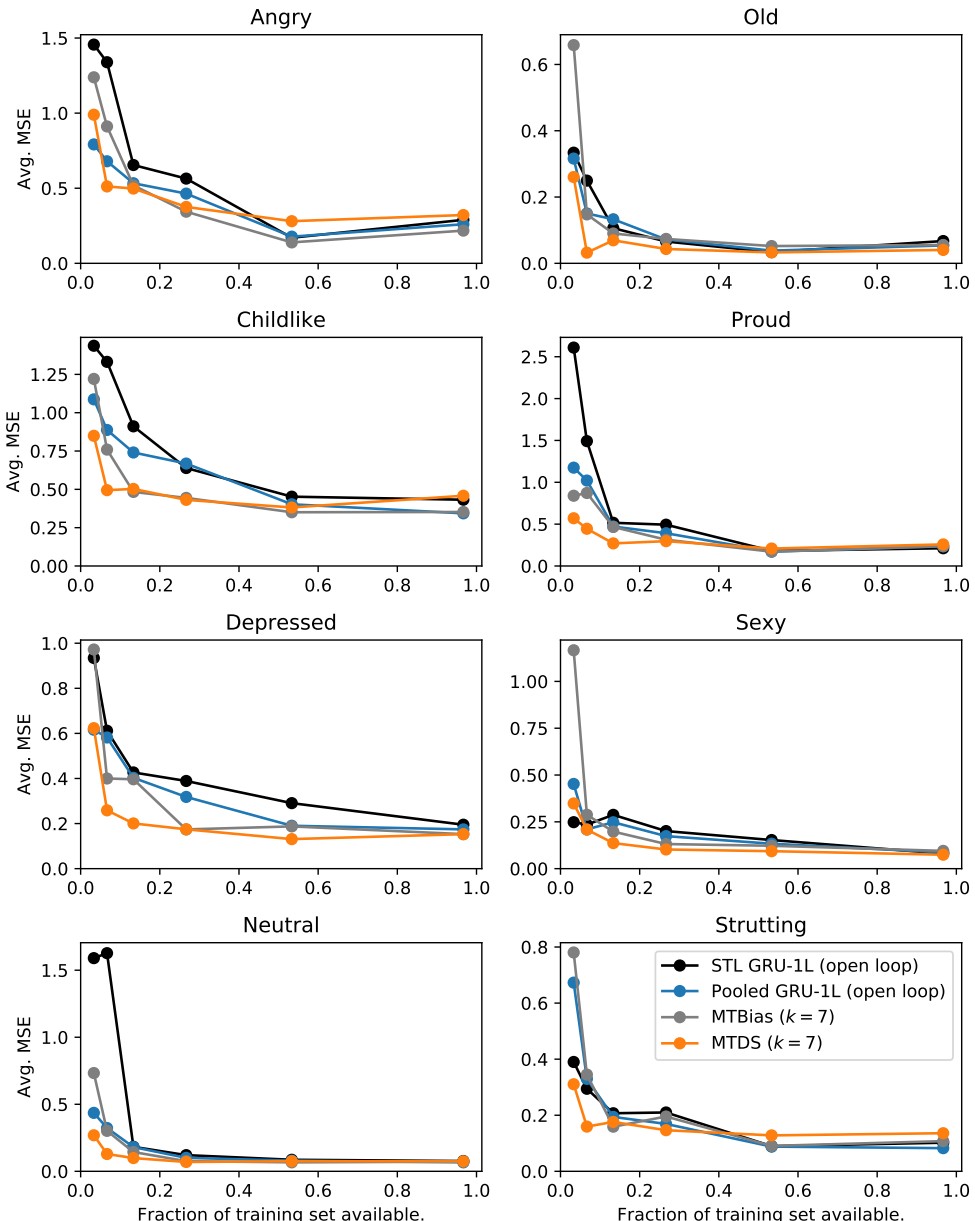

Figure 10: Per style MSE of Experiment 1.

| | **RMSE** | | | | | |
|---|---|---|---|---|---|---|
| **Model** | $t = 5$ | $t = 10$ | $t = 20$ | $t = 50$ | $t = 100$ | $t = 200$ |
| Training mean | 1.04 | 1.04 | 1.05 | 1.04 | 1.06 | 1.07 |
| Zero-velocity | 0.69 | 1.20 | 1.37 | 1.21 | 1.35 | 1.48 |
| 1-layer GRU (closed loop) | 0.35 | 0.64 | 0.81 | 1.00 | 1.45 | 7.28 |
| 2-layer GRU (closed loop) | **0.34** | 0.61 | 0.79 | 0.97 | 1.41 | 6.34 |
| 1-layer GRU (open loop) | 0.56 | 0.56 | 0.60 | 0.73 | 0.83 | 0.92 |
| 2-layer GRU (open loop) | 0.53 | 0.55 | 0.59 | 0.73 | 0.85 | 0.94 |
| MT Bias ($k = 3$) | 0.60 | 0.60 | 0.58 | 0.59 | 0.64 | 0.63 |
| MT Bias ($k = 7$) | 0.50 | 0.48 | 0.53 | 0.57 | 0.55 | 0.63 |
| MTDS ($k = 3$) | 0.61 | 0.62 | 0.59 | 0.61 | 0.63 | 0.63 |
| MTDS ($k = 7$) | 0.49 | **0.46** | **0.50** | **0.54** | **0.53** | **0.61** |

Table 7: Mocap Experiment 2 (novel sequences): average predictive MSE at $t = 5, 10, 20, 50, 100, 200$.

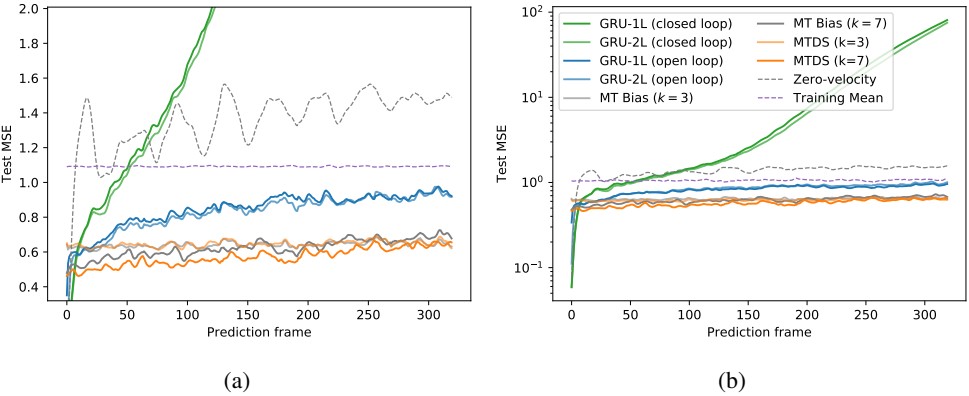

(a)
(b)

Figure 11: Results for Experiment 2 for all models on *(a)* truncated scale, *(b)* log scale.

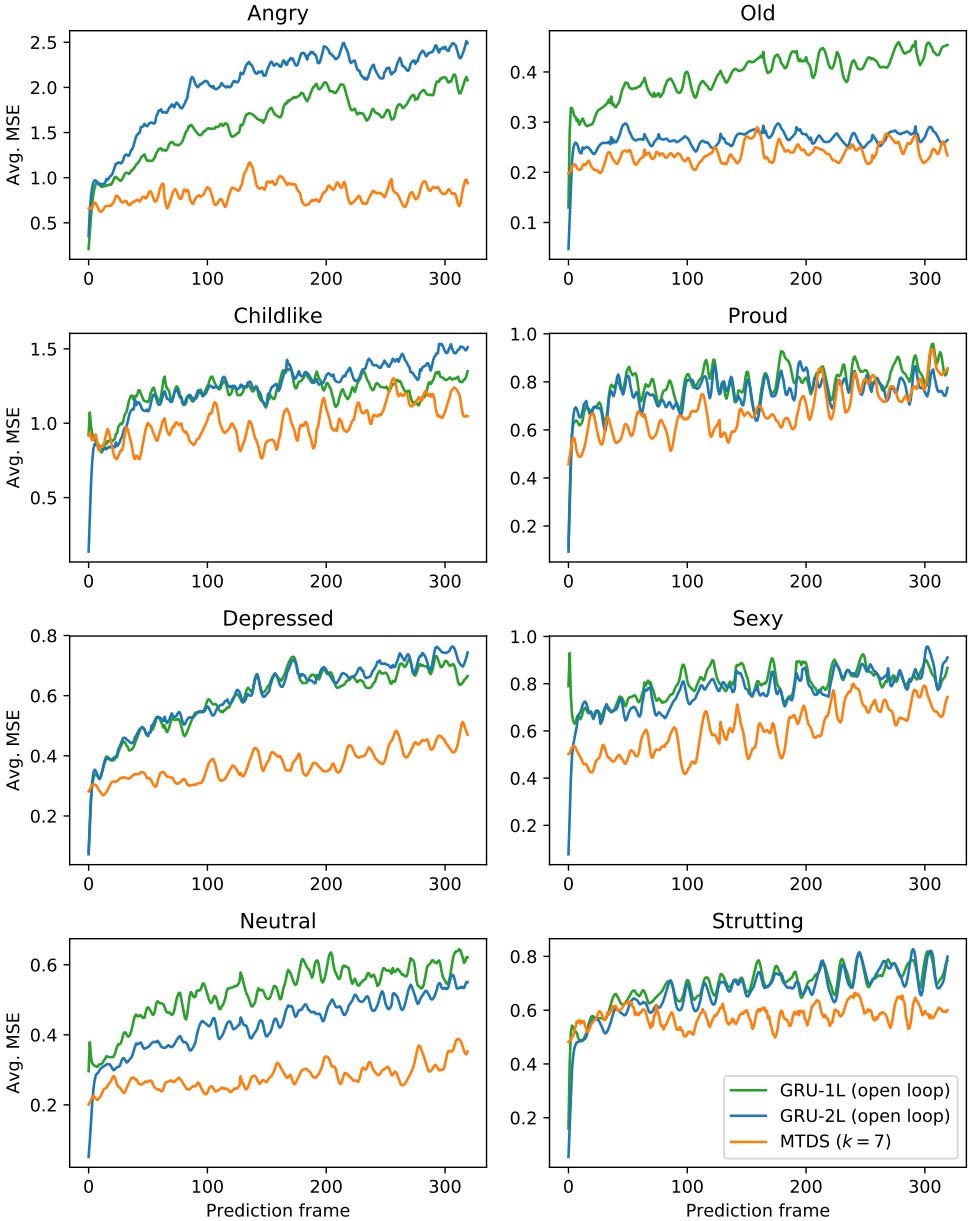

Figure 12: Per style MSE of Experiment 2.

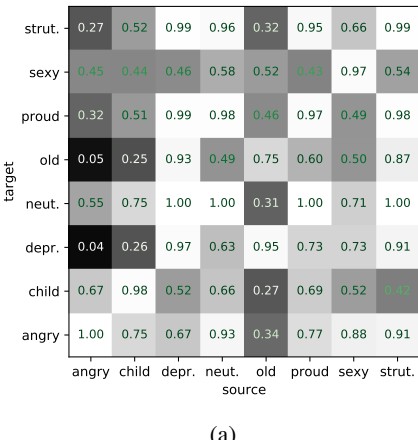 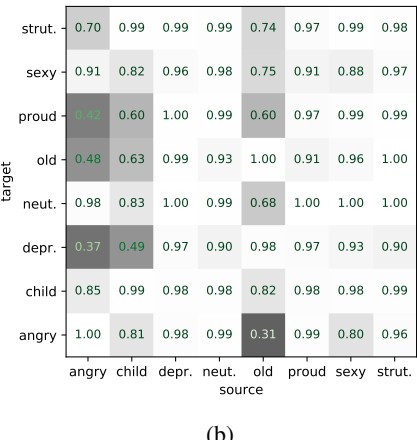

|  | (a) | | | | | | | | | | (b) |

Figure 13: Average classification accuracy for style transfer using inputs from source style (columns) and latent code $\mathbf{z}$ from target style (rows). There is no style transfer on the diagonal. *(a)* Results for model with MT bias only. *(b)* Results for full MTDS model.

The experimental setup is as follows. We carefully choose four segments of length 64 for each of the styles $s_1 = 1, \ldots, 8$ which represent the variability within each style. These correspond to the inputs $U_j^{(s_1)}$ for each source style $s_1$, with examples $j = 1, \ldots, 4$. We next seek the 'archetypal' latent code $\mathbf{z}$ associated with each target style $s_2$. For each $s_2$, we optimize $\mathbf{z}$ over 20 candidate values, obtained from the posterior mean of the style $s_2$ in the training set. Data are generated from all $\{U_j^{(s_1)}\}$ and the $\mathbf{z}$ which provides the greatest success in style transfer is chosen. The 32 highly varied input sequences guard against overfitting – the 'archetypal' codes for each style must perform well across much of the variety of the original dataset. We provide a scalar measurement of the 'success' of style transfer for each pair $(s_1, s_2)$ by using the resulting 'probability' that the classifier assigns the target style $s_2$, averaged across the four input sequences for the source $s_1$.

The results of these experiments are shown in Figure 13a for the model with multi-task bias, and Figure 13b shows the results for the full MTDS. Table 3c in the main text gives the marginal of these results wrt. the target style. The cells in Figure 13 give the classifier probability for the target style for each (source, target) combination, averaged over the four source inputs. Successful style transfer should result in a the classifier assigning a high score in every cell of the table. For most (source, target) pairs, the full MTDS model substantially outperforms the MTBias model: it appears that MTDS can control the prediction well in the majority of cases, and the MTBias model offers reduced control in general. However, we observe for both models that it is more difficult when styles are associated with extremes of the input distribution. Specifically, both the 'childlike' and 'angry' styles have unusually high speed inputs, and the 'old' style has unusually low speeds. Note that in order to provide style transfer, the models are mostly ignoring these correlations, even though they are very useful for prediction. Further improvements may be available, perhaps by using an adversarial loss, or applying domain knowledge to the model. This is orthogonal to our contribution, and we leave this to future work.

Providing style transfer from all varieties of source style is a challenging task. For instance, some styles include sources with widely varying speeds and actions, which may be mismatched to the target style. To understand what may be more typical use of the model, we provide an easier variant of this experiment where only one example of each source style is provided, rather than four. Note nevertheless that the same $\mathbf{z}^{(s_2)}$ is still used across all sources $s_1$. The results of this secondary experiment are provided in Figure 14. In this case, style transfer is successful for almost all (source, target) pairs in the case of the MTDS, except for the angry style. The MTBias model still has many notable failures.

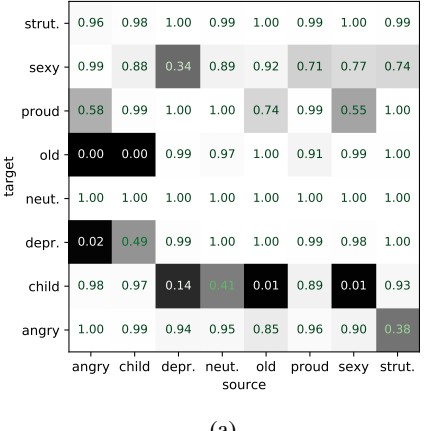
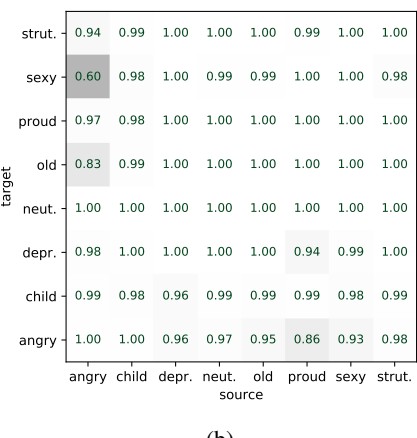

|  | (a) |  |  |  | (b) |
|---|---|---|---|---|---|

Figure 14: Average classification accuracy for style transfer where only a single source input is used for each (source, target) pair. The configuration of the matrix is the same as Figure 13. *(a)* Results for model with MT bias only. *(b)* Results for full MTDS model.

### A.4.1 GENERATED ANIMATIONS

A selection of animations are available online. We provide a link and a brief description of each set below. Where applicable, the animations show a comparison between the ground truth, the relevant MTDS model, and the (1 layer, open loop) pooled-GRU model. The latter was chosen by virtue of being the best competitor model in all experiments. In all cases, animations are a complete predictive rollout with no access to the ground truth.

1. **In-sample predictions** https://vimeo.com/362069486. The goal is to showcase the best possible performance of the models by predicting from inputs in the training set.

2. **MTL examples** https://vimeo.com/362122944. Examples from Experiment 1. We compare the quality of animations and fit to the ground truth for two limited training set sizes (6.7% and 13.3% of the full data). For both models, MSE to the ground truth is given, averaged over the entire predictive window (length 256). This is different to the experimental setup which uses only the first 64 frames.

3. **Novel test examples** https://vimeo.com/362068342. Examples from Experiment 2. We show the adaptions obtained by each model to novel sequences, in particular showcasing examples of the pooled GRU models inferring suboptimal styles wrt. MSE. Again, MSE to the ground truth is given averaged over the predictive window (length 256).

4. **Style morphing** https://vimeo.com/361910646. This animation demonstrates the effect of changing the latent code over time. This also demonstrates style transfer and style interpolation from experiment 3.

For style morphing, we found it useful to fix the dynamical bias of the second layer (parameter $\mathbf{b}$ in eq. 7) wrt. $\mathbf{z}$ since it otherwise resulted in 'jumps' while interpolating between sequences. We speculate that shifting the bias induces bifurcations in the state space, whereas adapting the transition matrix allows for smooth interpolation.

