# OpenReview forum: "Customizing Sequence Generation with Multi-Task Dynamical Systems"
_ICLR.cc/2020/Conference — Reject_

### Official Review · AnonReviewer3 · 2019-10-21
**Official Blind Review #796**

**Rating:** 6

**Review:**

<Strengths>
+ This paper proposes a new dynamic model named hierarchical multi-task dynamical systems (MTDSs) as a latent sequence model that enables users to directly  control the output of data sequence via a latent code z.
+ The proposed model adopts the multi-task learning idea to represent each sequence in the training set. Another strength of the model is flexibility as different base models and latent models can be chosen. In experiments, it can achieve reasonable performance using hierarchical RNN as a base model and VAE for latent modeling.
+ The new formulation is well-explained in the thorough appendix.


<Weakness>
In my opinion, the major weakness of this paper is weak experiments, although many results of different types of tests are reported.

1. The experiments are carried out with a simple toy dataset DHO and MOCAP data. As done in previous related papers, some challenging video datasets can be used for more convincing evaluation.

2. The compared baseline models are too weak, given that no state-of-the-art model is compared.
- Throughout the papers, only single-task version and pooped version of the model are compared. It may not be surprising that the multi-task version is better than these two weak baselines.
 -In section 3, several latent sequential models are introduced, including Miladinovic et al. (2019),  Hsu et al. (2017), Yingzhen & Mandt (2018), Bird et al. (2019). I strongly recommend a couple of them may need to be implemented and investigated in some sets of experiments.

<Conclusion>
My initial rate is borderline with slightly leaning toward accept. I am willing to adjust my rate more favorably if more experiments are conducted as mentioned above.


**Experience Assessment:**

I have published one or two papers in this area.

**Review Assessment: Checking Correctness Of Derivations And Theory:**

I assessed the sensibility of the derivations and theory.

**Review Assessment: Checking Correctness Of Experiments:**

I carefully checked the experiments.

**Review Assessment: Thoroughness In Paper Reading:**

I read the paper thoroughly.

---

> ### Author Response · Authors · 2019-11-14
> **Response**
>
> Thank you for your useful comments and evaluation above. We agree that providing additional comparison models will strengthen our presentation. Please see the updated experiments section for more details of our additional comparison model implementations; direct responses to comments are given below.
>
>                                                                                        ***
>
> 1. "The experiments are carried out with a simple toy dataset DHO and MOCAP data. As done in previous related papers, some challenging video datasets can be used for more convincing evaluation."
>
> Our focus is more on traditional time series modelling (which we note is a respectable area of research). Mocap data, which has a rich structure and moderately high dimensionality, is a good demonstration of contributions to this area. Our aim is to provide a holistic view of the situations where using an MTDS framework is likely to be useful, and those where it may not be.
>
>                                                                                        ***
>
> 2. "The compared baseline models are too weak, given that no state-of-the-art model is compared."
>
> We believe the baselines for the mocap data are fairly strong: the pooled GRU model is close to state of the art in the mocap literature [1]. In traditional benchmarks, this pooled approach cannot easily be beaten, but we consider different quantitative benchmarks that demonstrate the importance of explicit sequence customization. Various models proposed recently in the disentangled literature might also be co-opted to our context (these are typically concerned with demonstrating disentanglement, perhaps from large amounts of data, e.g. Miladinović et al., 2019). Our goal of exploiting customization is a complementary viewpoint.
>
> We have updated the introduction to give more prominence to this recent work. Importantly, we have also performed new experiments with a (deterministic state) version of Miladinović et al.: this performs poorly for customizing LDS models, but fairly well for RNNs. Nevertheless, our MTDS approach is superior in terms of data efficiency and user control, more details can be found in the updated version of the paper. We have decided against implementing Yingzhen et al. (2018) since it is unable to customize dynamics, and Hsu et al. (2018) would effectively result in the same model as Miladinović et al..
>
>                                                                                        ***
>
> Footnotes:
> [1]: As stated in the paper, this follows Martinez et al., 2017. As far as we are aware, the current state of the art is held by Pavllo et al. 2018, via the use of various data augmentation and pre-processing tricks, as well as possibly use of quaternion arithmetic in the RNN. The marginal gains are not large, and the differences are orthogonal to our work.
>
> References:
> * Hsu et al. (2018), Unsupervised Learning of Disentangled and Interpretable Representations from Sequential Data, NeurIPS 2017.
> * Martinez et al. (2017), On Human Motion Prediction Using Recurrent Neural Networks, CVPR 2017.
> * Miladinović et al. (2019), Disentangled State Space Representations, ICLR 2019.
> * Pavllo et al. (2018), Quaternet: A Quaternion-Based Recurrent Model for Human Motion, BMVC 2018.
> * Yingzhen et al. (2018), Disentangled Sequential Autoencoder, ICML 2018.

---

### Official Review · AnonReviewer2 · 2019-10-25
**Official Blind Review #2**

**Rating:** 6

**Review:**

This paper proposes a multi-task dynamical system for sequence generation. The model learns a number of parameters that represents the latent code z. The learned model can generate the customized individual data sequence and provide the smooth interpolation in the sequence space. The experiments on the synthetic data and the ocap dataset show the customization is beneficial for prediction tasks and enables for style transfer and morphing within generated sequences.

The mathematical definition and the proof in this paper are well justified. However, I had a hard time understanding the contribution of this paper.

- The main motivation of this paper is to treat each sequence as a task in the training set (customization of the individual data sequence). What are the advantages of this setting for sequence generation?

- Separate parameterization of the latent variable z for different tasks seems to be a key idea in this paper. What is the main benefit of parameterizing each latent variable? Does it improve diversity?

- If not, what is the key idea that the proposed model can generate diverse sequences compare to other generative models (e.g., GAN or VAE-based models).

- The authors claim that the proposed approach provides grater data efficiency. How is it compare to other generative models?

- Overall, it is not clear the benefit of the proposed model over existing algorithms. This paper needs to provide comparisons with any other existing models, especially on Mocap data,

- What are the differences between pooled models and single-task models?

----
I increase the rating after rebuttal, since the idea is interesting and the paper has been improved. However, I still think the experiments and the comparisons are unconvincing.

**Experience Assessment:**

I have published one or two papers in this area.

**Review Assessment: Checking Correctness Of Derivations And Theory:**

I assessed the sensibility of the derivations and theory.

**Review Assessment: Checking Correctness Of Experiments:**

I assessed the sensibility of the experiments.

**Review Assessment: Thoroughness In Paper Reading:**

I read the paper thoroughly.

---

> ### Author Response · Authors · 2019-11-14
> **Response**
>
> Thank you for your comments and questions above. In particular, it has been helpful in revising our introduction. We hope that the below helps to clarify these points, please let us know otherwise.
>
>                           ***
>
> 1. "The main motivation of this paper is to treat each sequence as a task in the training set.. What are the advantages of this setting for sequence generation?" ... "What is the main benefit of parameterizing each latent variable? Does it improve diversity?"
>
> We list some of the benefits of this approach (for example) at the end of para. 5 on page 1. Our objective is to provide explicit customization of dynamical systems predictions: (i) for simple dynamical systems where no customization is otherwise available, (ii) improving the scope where it is, and avoiding mode drift (sec. 1, para. 2) and (iii) in either case, permitting user control of this customization. For an example of our motivation, consider an animator controlling the style of mocap predictions from an RNN (e.g. a 'depressed' or 'childlike' walking style). Our approach is to learn a latent manifold over the dynamical system _parameters_ which can generate these different styles. This can be viewed as a form of multi-task learning (MTL), treating each sequence as a 'task', and learning different parameter settings for each. We agree that the motivation for MTL can be made a little clearer, we have updated the introduction in our revised version accordingly.
>
>                           ***
>
> 2. "[W]hat is the key idea that the proposed model can generate diverse sequences compare to other generative models (e.g., GAN or VAE-based models)."
>
> Our work here pertains to arbitrary-length sequences (such as in the time series context) rather than fixed-length sequences. Hence standard VAE- and GAN-based architectures are inappropriate. In contrast to generative autoregressive models such as NADE (Larochelle et al., 2011), and more recent forms such as PixelRNN (van den Oord et al., 2016), the key idea is that we can _control_ the style of the generation via a latent variable. Generative (dynamical) models which utilize ideas from VAE- and GAN-based architectures are given in the related work (section 3).
>
>                           ***
>
> 3. "The authors claim that the proposed approach provides grater data efficiency. How is it compare to other generative models?"
>
> Given our discussion of generative models above, two sensible comparisons will be to (i) a 'generative' RNN (generative in the same sense of e.g. a PixelRNN) and (ii) a recent VAE-based architecture given in Miladinović et al. (2019). A comparison to (i) is given by the 'Pooled-GRU' model -- where all the 'tasks' in the training set are pooled together. In response to your question, and also to R3, we have run the experiments for (ii), which are now available in our updated manuscript. In particular see the Mocap experiments (sec 4.2, 'MTBias' model) especially Figure 3(a).
>
>                           ***
>
> 4. "Overall, it is not clear the benefit of the proposed model over existing algorithms. This paper needs to provide comparisons with any other existing models, especially on Mocap data."
>
> The pooled-GRU model which serves as our benchmark is due to Martinez et al. (2017), which is approximately state of the art for this problem. Our contribution is to show that we can provide fine-grained control of the customization (see style transfer task and animations), and also that this produces better quantitative results in smaller data regimes and under dataset shift, for which our experiments provide strong evidence. As noted above, we also now provide a comparison with the most similar recent work to ours (Miladinović et al.) which further supports our claim. We have also tightened up the introduction to make the benefits more clear.
>
>                           ***
>
> 5. "What are the differences between pooled models and single-task models?"
>
> See sec. 4.1 (Model) and sec. 4.2 (Experiment 1). As we state in the introduction, sequence data often consists of multiple similar tasks mixed together (e.g. walking styles in mocap data). A pooled model thus ignores the distinction between the multiple tasks and learns a single model for the entire collection. A single-task (ST) model learns from data associated only with a single task (e.g. learns only from one walking style in the mocap data) and hence a much smaller training set. One can think of MTL as finding the 'sweet spot' in between these extremes. A minor clarification has been made in sec. 4.2 for ST models in view of this question.
>
>                           ***
>
> References:
> * Larochelle et al. (2011), The Neural Autoregressive Distribution Estimator, ICML 2011.
> * Martinez et al. (2017), On Human Motion Prediction Using Recurrent Neural Networks, CVPR 2017.
> * Miladinović et al. (2019), Disentangled State Space Representations, ICLR 2019.
> * van den Oord et al. (2016), Pixel Recurrent Neural Networks, ICML 2016.

---

### Official Review · AnonReviewer1 · 2019-10-26
**Official Blind Review #1**

**Rating:** 6

**Review:**


This paper proposes to add a latent variable to a dynamical, thereby encoding the notion of "task", eg, in Mocap, the latent variable could encode the walking style in an unsupervised manner.
Furthermore, it is then possible to generate new series conditioning of chosen latent variables.

The paper is clear and well written, the idea is interesting and the experiments seem well designed and convincing.

I do have just a couple of clarifications:
- in sec 2.1.1, you say that the diagonal*orthongal parametrization is valid without any loss of generality. If I understood well, to get to this for, you would still need to multiply the input x_t by U^{-1}. Therefore, any MTDLS can be written in this simplified form, correct, for a given input x_{1:T}, the formulation using the diagonal is more restrictive I believe. U^{-1} x could be of this form, but not necessarily x.
- in 2.3, the estimation of the posterior using your method of AIS with MoG seem quite involved, even though you mention it converges fast. What happened for simpler methods of estimating the posterior, did you see a drop in performance, run-tine only?

Overall I think this is an interesting paper, however I am not familiar with all the related work.

**Experience Assessment:**

I do not know much about this area.

**Review Assessment: Checking Correctness Of Derivations And Theory:**

I assessed the sensibility of the derivations and theory.

**Review Assessment: Checking Correctness Of Experiments:**

I assessed the sensibility of the experiments.

**Review Assessment: Thoroughness In Paper Reading:**

I read the paper thoroughly.

---

> ### Author Response · Authors · 2019-11-14
> **Response**
>
> Thank you for your detailed comments above. We hope that our helps to clarify these points, but -- while we appreciate time is short -- please let us know if not.
>
>                                                                                        ***
>
> > (sec 2.1.1, parameterization of LDS): "...the diagonal*orthongal parametrization is valid without any loss of generality... I believe. U^{-1} x could be of this form, but not necessarily x"
>
> To clarify, our argument is that any LDS distribution over the $y_{1:T}$ may be obtained using the restricted parameterization of eq. (6). We are careful not to claim that eq. (4) is equivalent to eq. (6) in isolation. We have made a slight tweak to the wording in the main text, and an additional note in the supp. mat. to emphasise this point.
>
> As to the relationship between the state variables, the $x_{1:T}$ of eq. (6) are indeed obtained via the transformation $U^{-1}$ of the state variables in eq. (4), with $U$ the matrix of left singular vectors of $A$. ($B$ and $b$ must also be transformed similarly, cf. eq. (14).) Replacing $A$ in eq. (4) simply by a diagonal matrix is certainly a restriction; specifically it excludes any oscillations in the latent state, and hence in observation space. We note in passing that this reduced parameterization (diagonal*orthogonal) was crucial in practice to stabilize the learning of the MTLDS.
>
>                                                                                        ***
>
> > (sec 2.3, inference): "the estimation of the posterior using your method of AIS with MoG seem quite involved... What happened for simpler methods?"
>
> We provide a wide-ranging discussion of inference techniques in supp. mat. A.1.4-A.1.5 which argues that there are relatively few inference approaches that can be used for this problem. An HMC/Langevin-MCMC approach will require a large number of (serial) gradient evaluations via BPTT, and hence will be very slow, but can utilize existing tools (e.g. Stan, Pyro). Using AdaIS with a simple Gaussian proposal will suffer in higher dimensions and/or for multi-modal or highly non-Gaussian posteriors, leading to reduced performance. The MoG variation adds relatively little complexity both conceptually, and in run-time (since run-time is dominated by forward simulation), and hence is more generally useful than a simple Gaussian proposal. Furthermore (as noted in p17, bottom), we observed highly non-Gaussian posteriors for small $t$ even in the MTLDS experiments.

---

### Author Response · Authors · 2019-11-14
**Response to All Reviewers**

We thank the reviewers for their careful reading and helpful comments. In light of this, we have uploaded a new version, for which we summarize major changes below. Individual concerns of each reviewer are addressed in separate comments.

- Section 4.1, 4.2: An additional competitor model has been added to each experiment, following Miladinović et al. (2019).

- Section 4.2 Experiment 3: In order to facilitate comparison with this competitor model, a summary table is given of the style transfer experiments (Fig 3c), and the previous pairwise comparisons relegated to supp. mat. A.4 to save space. This is the harder variant of this experiment (cf. variant ii in previous supp. mat., section A.4, experiment 3), with the easier variant also relegated to the supplementary material (Fig 14).

- Section 1: We have rewritten the introduction to give more prominence to existing approaches, give better context to the MTL approach, and highlight the benefits and contributions more clearly.

---

### Decision · Program_Chairs · 2019-12-19

**Decision:**

Reject

**Comment:**

This work proposes a dynamical systems model to allow the user to better control sequence generation via the latent z. Reviewers all agreed the that the proposed method is quite interesting. However, reviewers also felt that current evaluations were weak and were ultimately unconvinced by the author rebuttal. I recommend the authors resubmit with a stronger set of experiments as suggested by Reviewers 2 and 3.